# Recent transposable element bursts are associated with the proximity to genes in a fungal plant pathogen

**Ursula Oggenfuss[¤], Daniel Croll** [ID] *

Laboratory of Evolutionary Genetics, Institute of Biology, University of Neuchâtel, Neuchâtel, Switzerland

¤ Current address: Department of Microbiology and Immunology, University of Minnesota Medical School, Minneapolis, Minnesota, United States of America
* daniel.croll@unine.ch

**Data Availability Statement:** The genome assembly and annotation for genome assemblies are available at the European Nucleotide Archive (http://www.ebi.ac.uk/ena) under the BioProject PRJEB33986. TE consensus sequences, raw

## Abstract

The activity of transposable elements (TEs) contributes significantly to pathogen genome evolution. TEs often destabilize genome integrity but may also confer adaptive variation in pathogenicity or resistance traits. De-repression of epigenetically silenced TEs often initiates bursts of transposition activity that may be counteracted by purifying selection and genome defenses. However, how these forces interact to determine the expansion routes of TEs within a pathogen species remains largely unknown. Here, we analyzed a set of 19 telomere-to-telomere genomes of the fungal wheat pathogen *Zymoseptoria tritici*. Phylogenetic reconstruction and ancestral state estimates of individual TE families revealed that TEs have undergone distinct activation and repression periods resulting in highly uneven copy numbers between genomes of the same species. Most TEs are clustered in gene poor niches, indicating strong purifying selection against insertions near coding sequences, or as a consequence of insertion site preferences. TE families with high copy numbers have low sequence divergence and strong signatures of defense mechanisms (*i.e.*, RIP). In contrast, small non-autonomous TEs (*i.e.*, MITEs) are less impacted by defense mechanisms and are often located in close proximity to genes. Individual TE families have experienced multiple distinct burst events that generated many nearly identical copies. We found that a *Copia* element burst was initiated from recent copies inserted substantially closer to genes compared to older copies. Overall, TE bursts tended to initiate from copies in GC-rich niches that escaped inactivation by genomic defenses. Our work shows how specific genomic environments features provide triggers for TE proliferation in pathogen genomes.

## Author summary

Transposable elements (TEs) are engines of evolution over short and long evolutionary time scales and have played crucial roles in pathogen evolution. The impacts of TEs are multifaceted, ranging from creating adaptive sequence variants, gene disruptions, chromosomal rearrangements or even triggers of genome expansions. As a defense, pathogen genomes have evolved sophisticated mechanisms to silence or mutate TEs. Pathogens

sequences and phylogenetic trees are available on Zenodo (https://zenodo.org/record/7344421).

**Funding:** The authors received no specific funding for this work.

**Competing interests:** The authors have declared that no competing interests exist.

have also benefited from TEs thanks to altered virulence genes and increased antifungal resistance. How TEs cope with genomic defenses and expand in genomes (*i.e.*, cause TE bursts) remains poorly understood though. We analyzed over a dozen high-quality genomes of a fungal wheat pathogen species, which has recently experienced TE reactivations. We reconstructed the evolutionary history of many TEs by building phylogenetic trees. Using this approach, we identified "invasion routes", *i.e.*, tracking TE copies that constitute the most likely ancestors of renewed activity of TEs (*i.e.*, a bursts). Our work showed that specific features, in particular the proximity to genes, were likely important drivers leading the reactivation of TEs.

## Introduction

Transposable elements (TEs) are mobile DNA sequences in the genome. TEs create novel insertions by duplication or relocation of existing copies in the genome. Active TEs can proliferate in the genome at a very rapid pace (*i.e.*, burst), leading to disruption of coding regions, increased ectopic recombination rates, chromosomal rearrangements and finally genome size expansion. In the absence of bursts, most TE families are expected to reach a plateau in copy numbers determined by a balance between insertion and deletion rates [1–3]. Environmental stress factors are often the trigger of TE de-repression, which can in turn lead to bursts creating new copies [4,5]. Burst of TE activity is often followed by the activation of genomic defenses [3]. Even though some TE activity can be beneficial and drive short term adaptation to new environments, the TE insertion dynamics can be highly deleterious [2,6,7]. To counter TE activity, defense mechanisms have evolved to reversibly inactivate (*i.e.*, silence) or irreversibly mutate TEs [8,9]. Defenses against TEs include histone modifications, cytosine methylation, small RNA based silencing or KRAB zinc finger based transcriptional silencing [10–13]. Some TEs have the ability to regulate their own expression through small RNAs [14]. In certain ascomycete fungi, TEs are additionally targeted by repeat-induced point mutations (RIP) during sexual recombination [15–17]. RIP has the potential to introduce CpA to TpA mutations in any copy of a duplicated sequence, leading to decreased GC contents and a loss-of-function risk at targeted loci. In the ascomycete *Neurospora crassa*, a few generations of sexual recombination are sufficient to degenerate TE copies [18]. RIP has reduced efficacy on sequences shorter than ~500 bp in *N. crassa* and has almost no impact on repeats below ~200 bp allowing small TEs to escape [19]. Defense mechanisms against TEs may be weakened under stress conditions or lost over evolutionary time scales [20–22]. Additionally to defenses against TEs, TE bursts are counterbalanced by deletion via ectopic recombination at the individual level, and via purifying selection and genetic drift at the population and species level [1,2,23].

The combination of TE activity, defense mechanisms and selection shape the genomic environment. Chromosomal sequences are often highly heterogeneous that contain niches of different levels of TE or gene density, or different chromatin states [24,25]. How TE insertions reshape the genomic landscape depends on the fitness effects of new insertions [26]. Generally, strong purifying selection acts against most new insertions in plant, animal and fungal species [27–31]. Yet, the genomic environment can differ dramatically between species. Many yeast species with extremely low repeat content carry genomes with generally only gene-dense niches [32,33]. In contrast, maize carries a very large repeat-rich genome, with a low gene content and highly isolated genes [34]. New TEs are more likely to insert into a coding region and have deleterious effects in compact yeast genomes, compared to maize, where a novel TE most likely will insert into other TEs creating nested insertions. Such distinct properties of genomes

 

can be framed as "resistance" or "tolerance" towards TE insertions [35]. Genome compartmentalization in plant pathogens is also referred to as the *two-speed genome* with the tight links between detrimental and beneficial effects of TE insertions being framed as a *Devil's Bargain* [35–39]. In *two-speed genomes*, genomic niches are well pronounced, dividing gene-rich, repeat-poor niches, and repeat-rich niches that harbor genes associated with the interaction with the plant host. Rapid diversification of such genes can be driven by the genomic environment, in particular TEs [37,40,41]. Finally, the epigenetic landscape diversifies genomic niches into transcriptionally open euchromatin (H3K4me) and heterochromatin niches tightly packed around histones. Generally, TEs are unable to insert in heterochromatic regions nor are TEs transcriptionally active [42]. Facultative (H3K27me3) marks many TE-dense regions, which can be de-repressed during stress in fungal plant pathogens (*e.g.*, induced by stress during infection), potentially leading to TE activation [5,43]. Other regions with heterochromatin marks (constitutive heterochromatin; H3K9me3) remain repressed during stress conditions, and TEs will not be activated [5,44].

TEs are highly diverse in terms of length, coding regions or transposition mechanism. Generally, TEs are clustered into retrotransposons and DNA transposons, each with a number of orders and superfamilies [45,46]. Retrotransposons create new copies via an RNA intermediate and a *copy-and-paste* mechanism, and are separated into elements containing long terminal repeats (LTRs, *e.g.*, *Copia* or *Ty3*) and non-LTR (*e.g.*, LINEs) [45]. DNA transposons are separated into DNA transposons that use a *cut-and-paste* mechanism and contain terminal inverted repeats, and Helitrons with a *peel-and-paste* mechanism [45,47]. Some TEs, including MITEs (miniature inverted repeat transposable elements) lost their coding region and rely on the transposition mechanism of full-length TEs. At the level of sequence similarity, TEs can further be grouped into TE families. The activation and burst of a TE family will create a high number of novel TE insertions that are exact copies, that only diverge slowly with time [48]. Hence, phylogenetic analysis of all TE copies of a family can be used to reconstruct the evolutionary history of the TE family. Analogous to viral birth-death models, bursts of transpositions should leave distinct marks of short internal branches in phylogenetic trees [49,50]. Transposition bursts are characterized by a most recent common ancestor likely reflecting the copy initiating the expansion. In contrast, copies with long terminal branches have likely been silenced or mutated via RIP. TEs with long terminal branches have likely lost functionality, will initially remain in populations as remnants, accumulate mutations and ultimately degenerate. Reconstructing the succession of events leading to transposition bursts is often challenged by the difficulty in recovering all copies of a TE within a given species. The difficulty stems from the incomplete nature of many genome assemblies and the fact that TE copies are not fixed in populations. Recovering full-length copies of TEs that resulted from transposition bursts remains challenging without high-quality genome assemblies. Additionally, individual genomes of a species typically carry only a small subset of all TE copies present within the species.

*Zymoseptoria tritici* (previously *Mycosphaerella graminicola*) is an important fungal plant pathogen on wheat that shares an extended history of co-evolution with its host [51]. TEs cover 16.5–24% of the genome, often located in repeat-dense niches with genes involved in the interaction with the host [52]. *Z. tritici* has a moderate TE content, and recent TE activity has produced both beneficial insertions and negative impacts on the genome structure. TE activity is likely associated with incipient genome size expansions [31] and the emergence of adaptive traits. For example, different TEs have inserted in the promoter region of a major facilitator superfamily transporter gene and facilitated multidrug resistance in *Z. tritici* [53–55]. Furthermore, increased activity of a DNA transposon reduced asexual spore production, melanization and virulence [56–58]. Cytosine methylation was lost after a duplication event of the DNA

methyltransferase *MgDNMT* gene because subsequent RIP mutations rendered all copies non-functional [59,60]. Some populations of the pathogen have likely lost *dim2*, an essential gene of the RIP machinery [61,62]. In contrast to older copies, a subset of young TEs show no apparent RIP signatures in recently established populations outside of the center of origin [63]. The loss of RIP seems to be a gradual process accompanying the global spread of the pathogen. Histone modifications and small RNAs likely contribute to silencing of TEs [64–66]. The TE repertoire includes 304 TE families based on an analysis of 19 completely assembled genomes [52]. A subset of TEs show evidence for de-repression during plant infection likely connected to bursts of TE proliferation [5,31,52].

Here, we propose a pangenome-based approach to reconstruct the recent evolutionary history of TEs in *Z. tritici*. We retraced the invasion dynamics of young TEs using phylogenetics to determine triggers of TE expansions. We established near total evidence for all copies of multiple TE families by gathering information from 19 telomere-to-telomere genomes combined into a pangenome. We used phylogenetic reconstruction of ancestral TE states to identify genomic niches of TE activation and proliferation. The broad view of TE invasion routes suggests that TE copies near genes act as triggers for copy number expansions. Escape from genomic defenses is a likely the major driver of TE dynamics.

## Results

### Transposable element diversity within *Z. Tritici*

We analyzed 19 chromosome-level genomes to comprehensively map the genome-wide distribution of TE families in the fungal wheat pathogen *Z. tritici* (Fig 1A; [52]). Genome size and TE content vary considerably among individuals and show a positive correlation (r = 0.78, p < 0.001) [52]. We grouped TEs into MITEs, RLC/RLG, LINE and others (Fig 1B) with consensus sequences and individual TE copies accessible on Zenodo (https://zenodo.org/record/7344421). The genomes of an Australian and Iranian isolate have the highest TE content (24%) and lowest TE content (16.5%), as well as the largest genome (41.76 Mb) and smallest genome (37.13 Mb), respectively (Fig 1C and S1 Table). We focused on TE families with at least 20 copies, which collectively have a total of 23,395 copies across all analyzed genomes. Around half of the copies belong to DNA transposons (*n* = 10,586 copies in 104 families), half to retrotransposons (*n* = 11,907 copies in 59 families; S2 Table) and only few remained unclassified (*n* = 902 copies in seven families) without meaningful differences among genomes (S2 Table and S1 Fig). We compared frequencies of locus specific TE copies between the 19 genomes, and found most TE copies to be singletons or at low frequency (Fig 1D). Only few TE loci were fixed, (*n* = 122; Fig 1D) and these predominantly belong to MITEs.

### Niches of transposable elements have low gene content

We assessed variation in the genomic environment in 5 kb windows both up- and downstream of each TE copy (Fig 2A). For the reference genome IPO323, we found no accumulation of TE copies in niches with marks of open chromatin (*i.e.*, euchromatin; H3K4me9). A small subset of TE copies overlaps with constitutive heterochromatin marks (H3K9me3), or facultative heterochromatin (H3K27me3) (Fig 2A). Across all 19 genomes, we found that most TEs are located on core chromosomes (*i.e.*, chromosomes shared among all isolates), but TE copies are at higher density on accessory chromosomes (Fig 2B). The majority of TE copies are located in niches with a GC content below 50%, with the exception of MITEs (average of 72.0% GC; Fig 2B). Only a small subset of TE copies is in niches overlapping a gene or a subtelomeric region (Fig 2B). We found no overall association between TE copies and large RIP-affected regions. However, most RLC, RLG and LINEs are located inside and most MITEs outside of large RIP affected regions (Fig 2B).

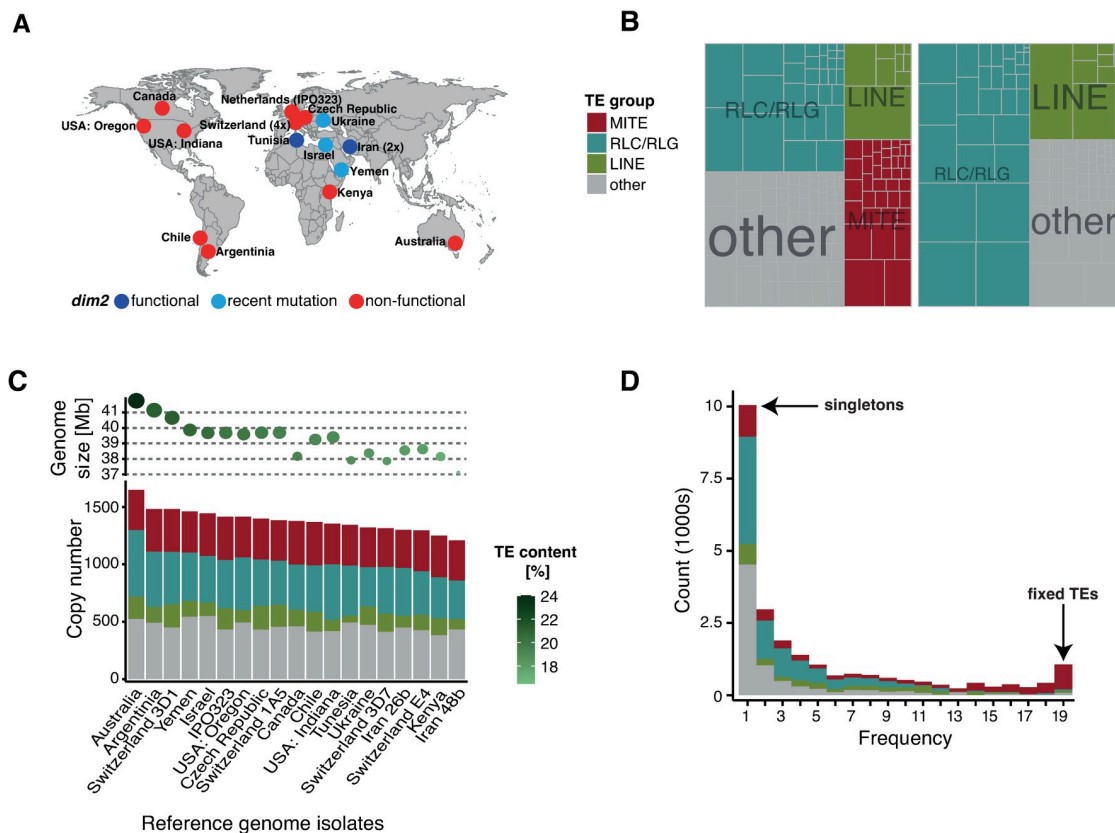

**Fig 1. Transposable element (TE) distribution in 19 telomere-to-telomere genomes of *Zymoseptoria tritici*:** (A) Origin of reference genome isolates originally used for the *Z. tritici* pangenome reported by Badet et al [52]. The color indicates the status of the *dim2* gene with an important influence on RIP. Data from Möller et al [62]: dark blue = present and functional, bright blue = recently mutated, red = non-functional. Map created with *ggmap* version 3.0.0 [73]. Map data from OpenStreetMap: https://www.openstreetmap.org/copyright. (B) Genome size and TE copy number per isolate. Circle sizes indicate the genome size, the green shade indicates the TE content. The colors indicate MITEs (miniature inverted repeat transposable elements, small non-autonomous DNA transposons corresponding to several TE superfamilies), RLC and RLG (two superfamilies belonging to LTR) and LINE. (C) Copy numbers of TEs (left) and total length (right) in all 19 genomes. Smaller boxes correspond to TE families. (D) Allele frequency distribution of TEs at orthologous loci among genomes. TEs were defined as orthologous if they were located between the same set of orthologous genes.

More than one third of TEs are inserted into niches with more than 80% TE content. In contrast, MITEs are preferentially inserted into TE poor niches (Fig 3A). GC content in TE copy niches varies between 25–60%. We found more than one third of TE copies 1–10 kb away from the next gene, with MITEs being on average closer (Fig 3B). TE copies are often close to RLC (*Copia/Ty1*), RLG (*Ty3*, formerly also *Gypsy*, see [67]) and LINE copies (902 and 2431 bp average distance, respectively). MITEs generally are at a distance 8,037 bp from the next TE on average (Fig 3C). Overall, TE density is negatively correlated with gene density and GC content (Fig 3D). TEs and TE fragments belonging to families with longer consensus sequences tend to be located in already TE rich niches. RIP and GC content are strongly negatively correlated, yet low GC content is unlikely explained by RIP alone (Fig 3D).

## Recent activity of high-copy transposable element families

Recently active TE families typically carry a high number of similar TE copies in the genome. We first filtered for a subset of TE families with more than 100 copies in all 19 genomes

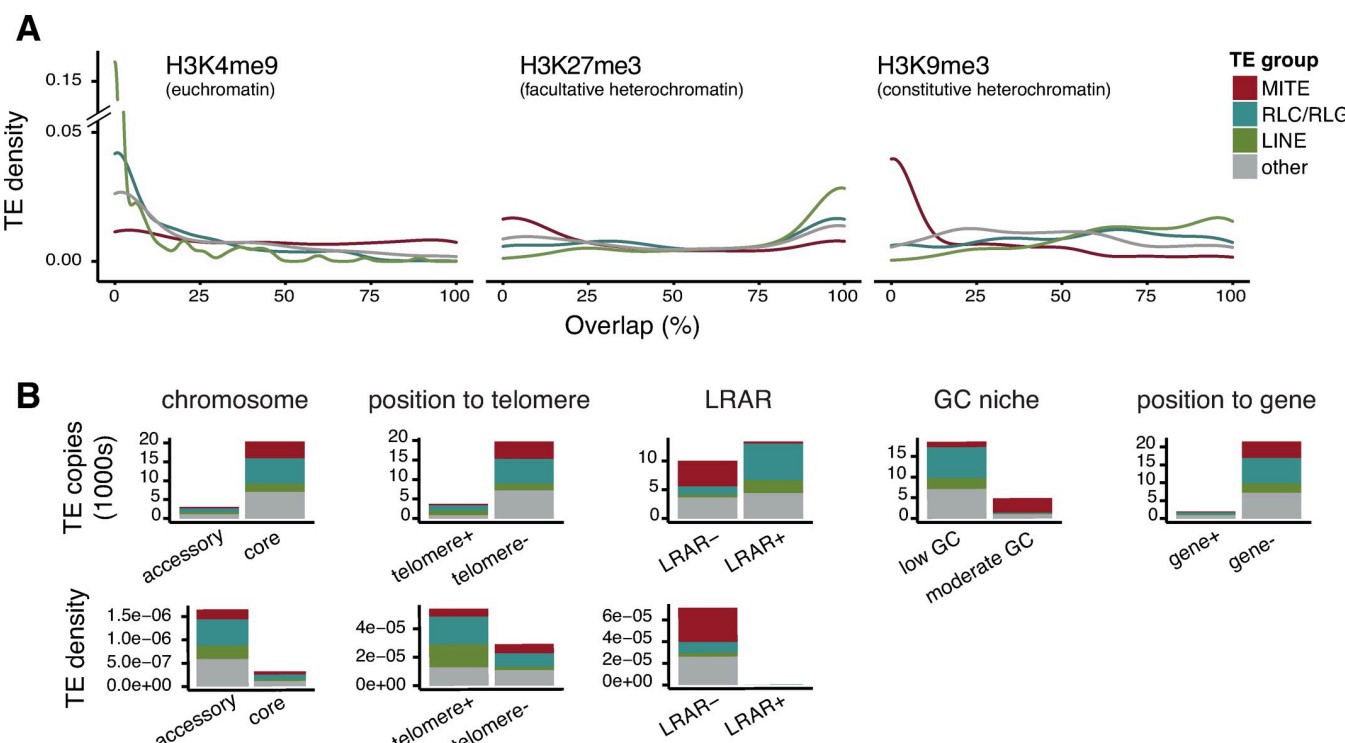

**Fig 2. Characteristics of TE niches across the genome**: (A) Proportional overlap of H3K27me2, H3K4me9 and H3K9me3 histone methylation marks with TE niches in the reference genome IPO323. Colors indicate the group of TE. (B) TE copy numbers between core and accessory chromosomes (copy number and density): in and outside the subtelomeric region (copies and density); into large RIP affected regions (LRAR; copy and density); into niches with a moderate ($\geq$ 50%) or low (< 50%) GC content; overlapping regions annotated as genes.

combined. The 61 retained TE families predominantly include MITEs ($n$ = 12) as well as RLC and RLG ($n$ = 5 and 11, respectively; S2A Fig). We find that high-copy TE families tend to also have more variable copy numbers among 19 genomes (Fig 4A). The GC content of high copy TE families generally is lower than 50%, with the exception of MITEs that have an overall higher GC content around 52%, and an extreme case with the RLC_Deimos family where most copies have a GC content below 40% (Fig 4B). Full-length TE copies range from 218–13,907 bp, with the shorter copies belonging to the non-autonomous MITEs lacking coding regions and the longest copies belonging to RLG, RLC, DHH and RIX (Fig 4C). To estimate the genetic distance of TE copies, we calculated the nucleotide diversity for each TE family. RIP activity on TE copies is expected to increase genetic distances among copies. TE families with the highest copy numbers showed lower nucleotide diversity consistent with recent proliferation in the genome, and despite having stronger signatures of RIP. MITEs tend to have higher nucleotide diversity at similar copy numbers compared to other TE families (Fig 4D and 4F). MITEs are also less affected by RIP (Fig 4E and 4F). Terminal branch lengths of individual TE copies are a further indication of increased genetic distances. Copies of MITEs tend to have overall short terminal branch lengths compared to other TEs (S2B Fig). The short length of MITEs might constrain the potential to accumulate mutations compared to longer TEs. Consistent with this, many MITEs show long internal branch lengths between distinct clades characterizing independent bursts (S3 Fig). Overall, TE families with high copy numbers and long consensus sequences show lower nucleotide diversity, however, most of the mutations are RIP-like mutations (Fig 4F).

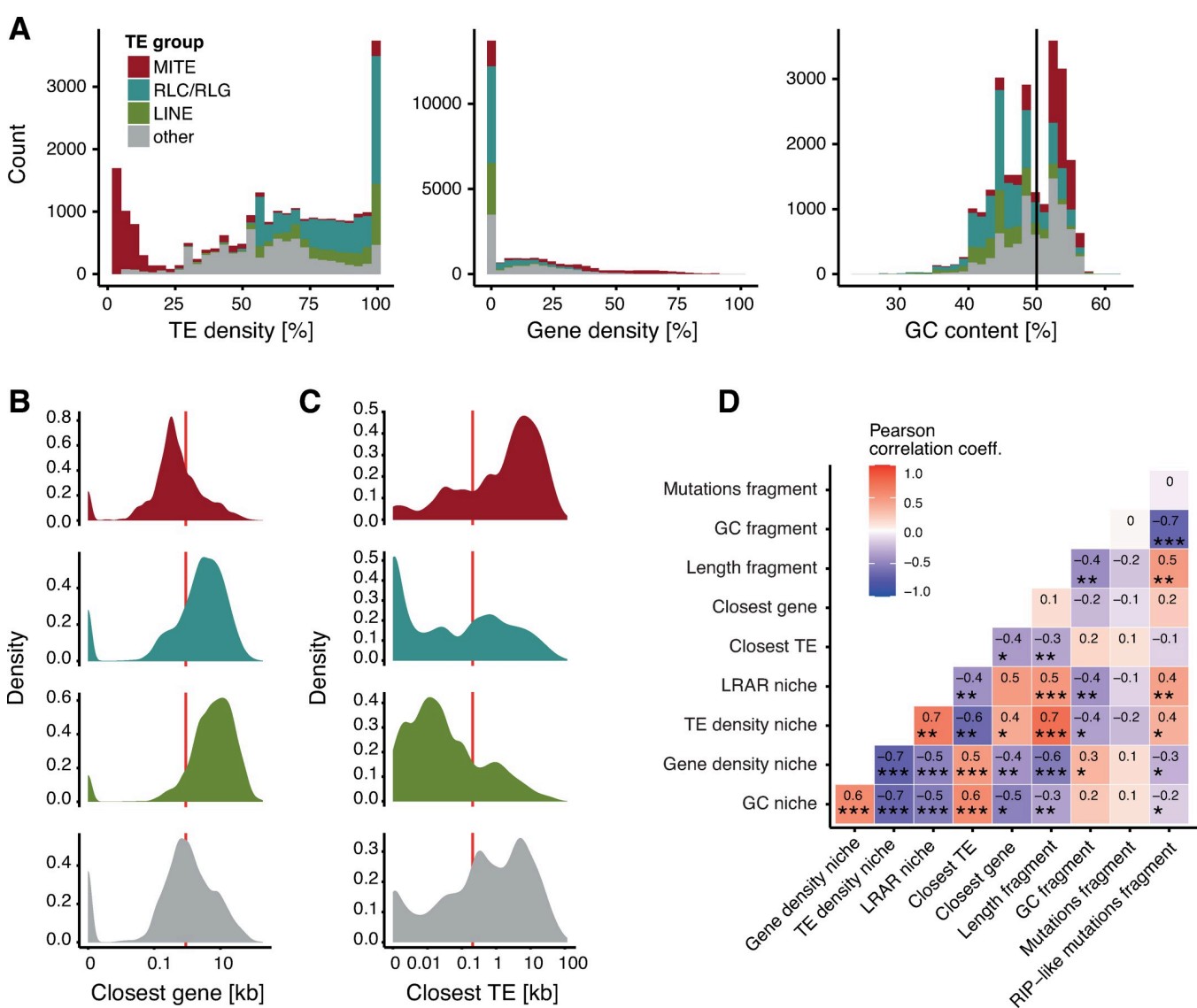

**Fig 3. Distribution of niche and TE copy characteristics**: (A) Overlap of TE content, gene content and GC content with TE copy niches. (B) Distribution of the distance to the closest gene in MITEs, RLC/RLG, LINE and other TEs. The red line indicates the mean distance. (C) Distances to the next TE MITEs, RLC/ RLG, LINE and other TEs. The red line indicates the mean distance. (D) Pearson correlation matrix of 11 characteristics of TE copy niches and TE copy characteristics. Dark red indicates strong positive correlation, dark blue indicates strong negative correlation of two characteristics. * p < 0.05, ** p < 0.01, *** p < 0.001.

### Transposable element expansion routes identify genomic niches of proliferation

To disentangle the factors influencing recent TE bursts, we first calculated genetic distances among copies (Fig 5A). Most TE families show their highest activity in a similar, recent age range. We found two TE families with ongoing activity (Styx, a IS3EU DNA transposon and RLX_LARD_Thrym). Among the high copy TE families, the RII_Cassini has been most recently active. RLG_Luna, RLG_Sol, RIX_Lucy and RLC_Deimos have undergone earlier bursts with both RLC_Deimos and RLG_Luna showing indications of multiple episodes of rapid proliferation. To reconstruct TE expansion routes in the genome, we built phylogenetic

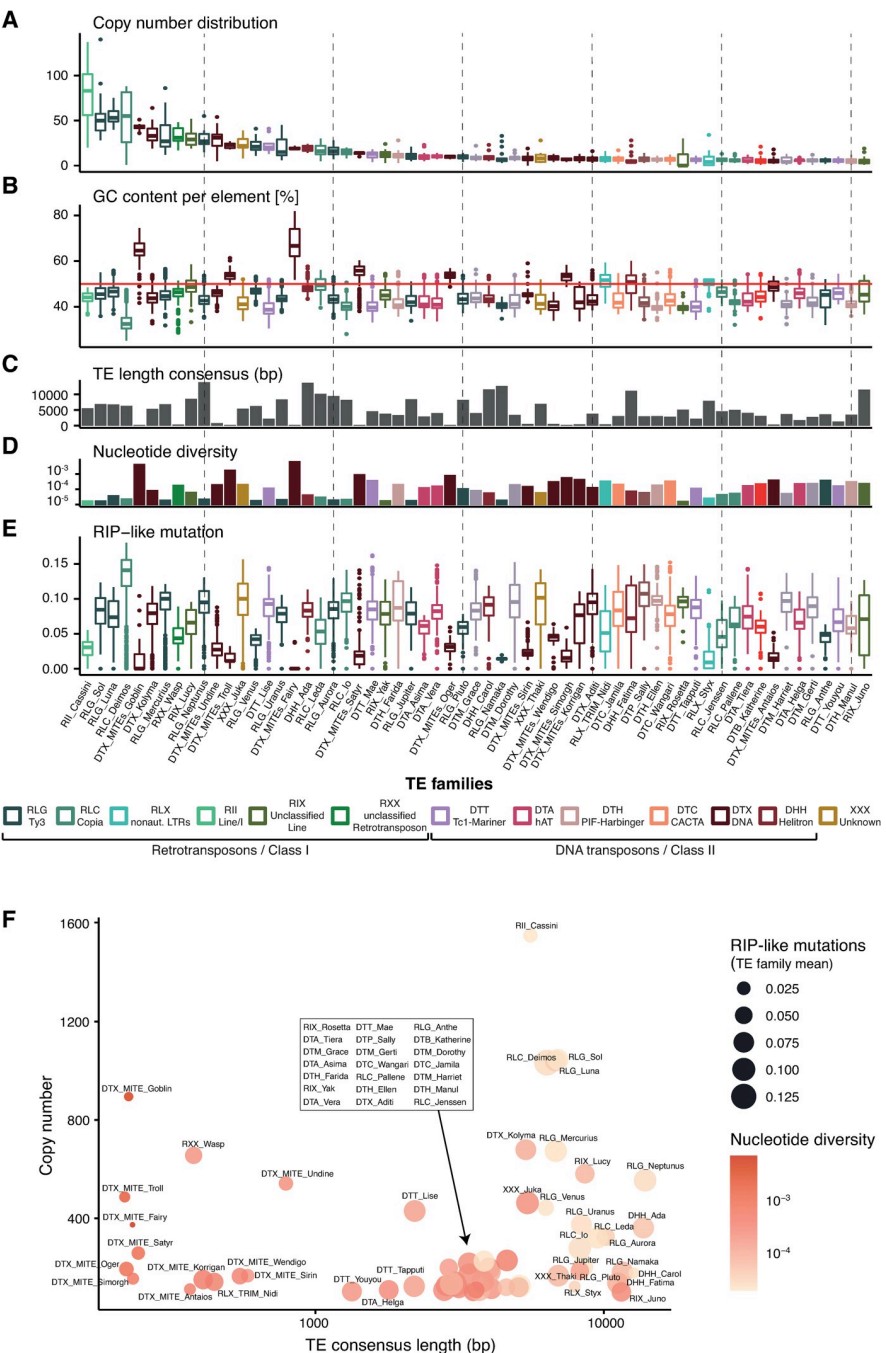

**Fig 4. Characteristics of high-copy TE families:** TE families are ordered from the highest copy numbers to lowest copy numbers (right) in all 19 analyzed genomes combined. (A) Distribution total copy number per TE family and isolate. (B) GC content distribution per TE family. The red line represents a GC content of 50%. (C) Length of the consensus sequence corresponding to the full-length consensus sequence excluding nested TEs or partial deletions. (D) Nucleotide diversity of the TE family (transformed as log10(nucleotide diversity*100,000)). (E) Number of RIP-like mutation (CpA<->TpA/TpG<->TpA) per TE copy, corrected for the length of the TE. (F) Correlation between copy numbers and consensus sequence lengths for TE families. Circle size corresponds to the mean number of RIP-like mutations and the color indicates the nucleotide diversity.

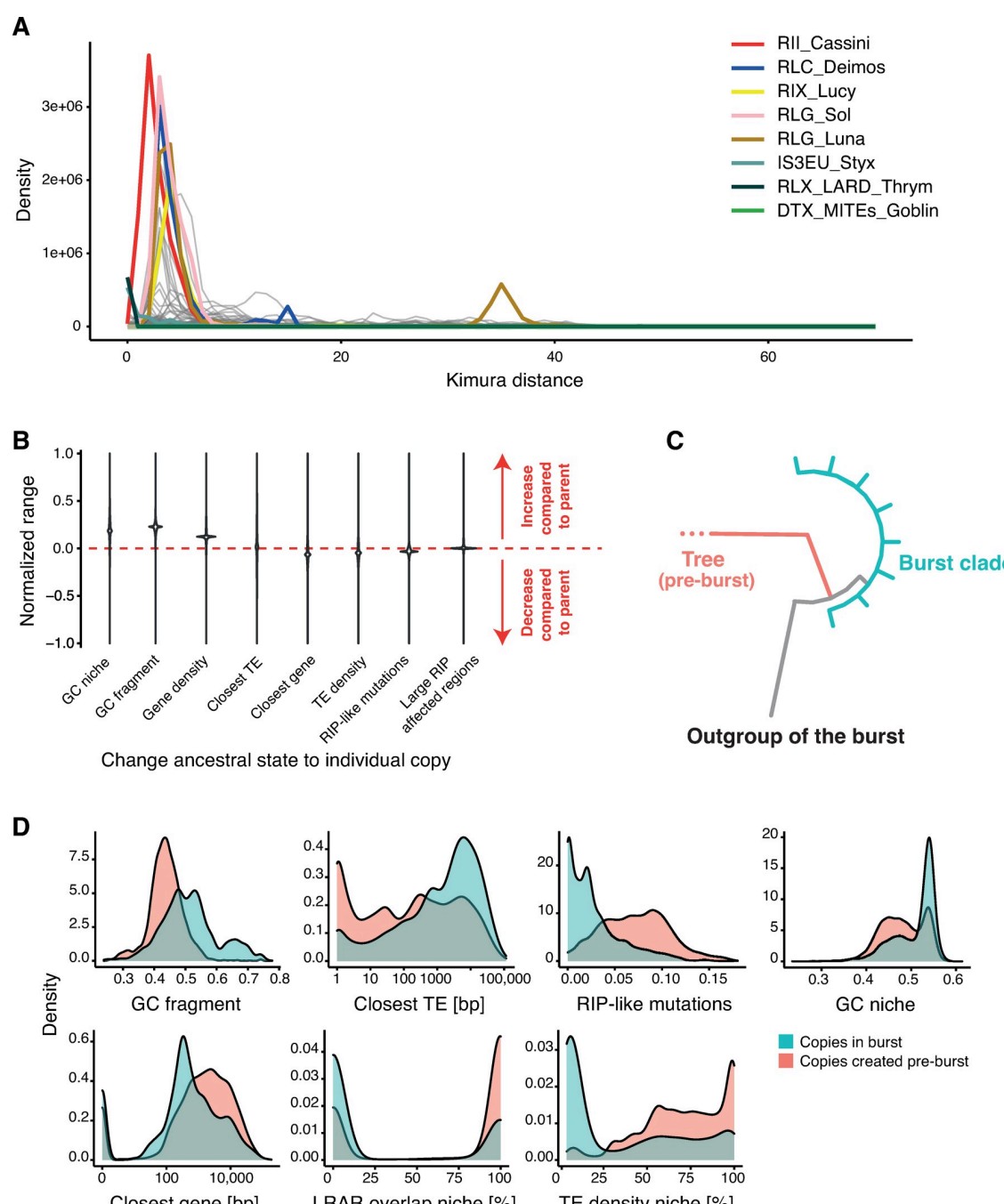

**Fig 5. Genomic origin and features of TE transposition bursts:** (A) Repeat landscape of the TE families with the highest copy numbers. Colors indicate the highest copy numbers (RII_Cassini, RLC_Deimos, RLG_Sol, RLG_Luna), TEs with multiple bursts (RLC_Deimos, RIX_Lucy and RLG_Luna) or very recent burst (Styx, Thrym). DTX_MITEs_Goblin is included but shows no apparent activity. (B) Normalized range of characteristics of TE copies and their genomic niche compared between the estimated ancestral states and derived copies. A positive value indicates that metrics increased compared to the ancestor sequence, and a negative value indicates a decrease. (C) Scheme of the definition of burst and burst outgroups based on phylogenetic trees of TE families. Green indicates the copies of a burst with low terminal branch lengths and the red outgroup indicates the closest related sister branch of a burst. (D) Distribution of niche and TE copy characteristics of copies belonging to a burst clade (blue) compared to all other copies (red).

trees (on Zenodo https://zenodo.org/record/7344421), rooted using TE copies found in *Zymoseptoria* sister species. Using tree reconstruction, we assessed for each TE copy whether characteristics of the TE sequence and the genomic niche evolved from the parental node in the tree. We used ancestral state reconstruction to identify niche and sequence features of all internal nodes on each family's tree. Compared to the ancestral state, we found increases in GC content of the TE sequence, as well as the GC and gene content of the genomic niche (Fig 5B). The distance to the closest gene decreased, and TEs are in regions with a lower TE density compared to their direct ancestors (Fig 5B). While most TE copies are located on a different chromosome compared to the parental node, new insertions typically remain on core chromosomes (65.4%) or switch from an accessory to a core chromosome (21.5%). We found that more than half of the TE copies remain in isochores of low GC content (58.4%) or jumped from moderate to low GC content (20.5%). Additionally, a large part of TE copies either remain in large RIP-affected regions or jumped into such regions (20.7%).

We identified individual bursts within TE families by retrieving clades of highly similar sequences distinct from other sequences in the tree (Fig 5C). We defined the outgroup of individual bursts as being closest to the parental copy that preceded the burst. Overall, around 50% of TE families experienced at least one recent burst and 10% (*n* = 32) of all TE families revealed several bursts. Copies resulting from individual bursts are often found only in a subset of the analyzed genomes of the species, consistent with TE bursts being recent. Yet, most burst are composed of copies originating from multiple genomes. In MITE families, a large proportion of all copies likely originate from a recent burst. To identify general properties of TE bursts, we compared copies created in a recent burst with all other copies of from that TE family (Fig 5D). Burst copies generally have a higher GC content, less RIP-like mutations, are closer to genes and are more distant to other TEs compared to non-burst copies (Fig 5D). We found also that burst copies tend to occupy genomic niches with lower TE density, overlap less likely with large RIP-affected regions and are located in niches with a higher GC content.

## Niche characteristics of transposable element activation

We focused on five TE families with particularly high copy numbers and evidence for large bursts to identify drivers of expansions (*i.e.*, LINE/*I* RII_Cassini, LTRs RLG_Luna RLG_Sol, and RLC_Deimos, and the MITE DTX_MITE_Goblin). Even though RLG_Luna and RLG_Sol both contain among the highest copy numbers, they show no indication of recent bursts (Fig 6). In comparing niche characteristics of older TE copies with copies generated during bursts, we detected no universal pattern shared by all TE families. There are indications that TEs part of a burst are in regions with a reduced impact of RIP showing generally higher GC content (Fig 6A), and being located in regions with a lower LRAR (Fig 6B). Nevertheless, there is no significant difference in the GC content of niches between older and burst TE copies (Fig 6C). Most TEs remain at a larger distance from genes, only copies of RLC_Deimos show a clear trend to be closer to genes for copies generated during a burst (Fig 6D). Analyzing the 38 RLC_Deimos copies that inserted into genes in the reference genomes of IPO323, three gene annotations (7.8%) encode *Copia*-like function, suggesting that a TE copy was wrongly annotated as a gene. Another eight genes (21.1%) lack any known function (including TE functions) and the remaining 27 genes (71.1%) have predicted protein functions unrelated to any known TE. Compared to the consensus sequence, TEs in a burst have a higher accumulation of mutations in RLC_Deimos (Fig 6E), yet the mutation rate increase seems not to stem from RIP-like mutations (Fig 6F). Other TE families share no similar pattern of temporal escape from RIP facilitating a burst.

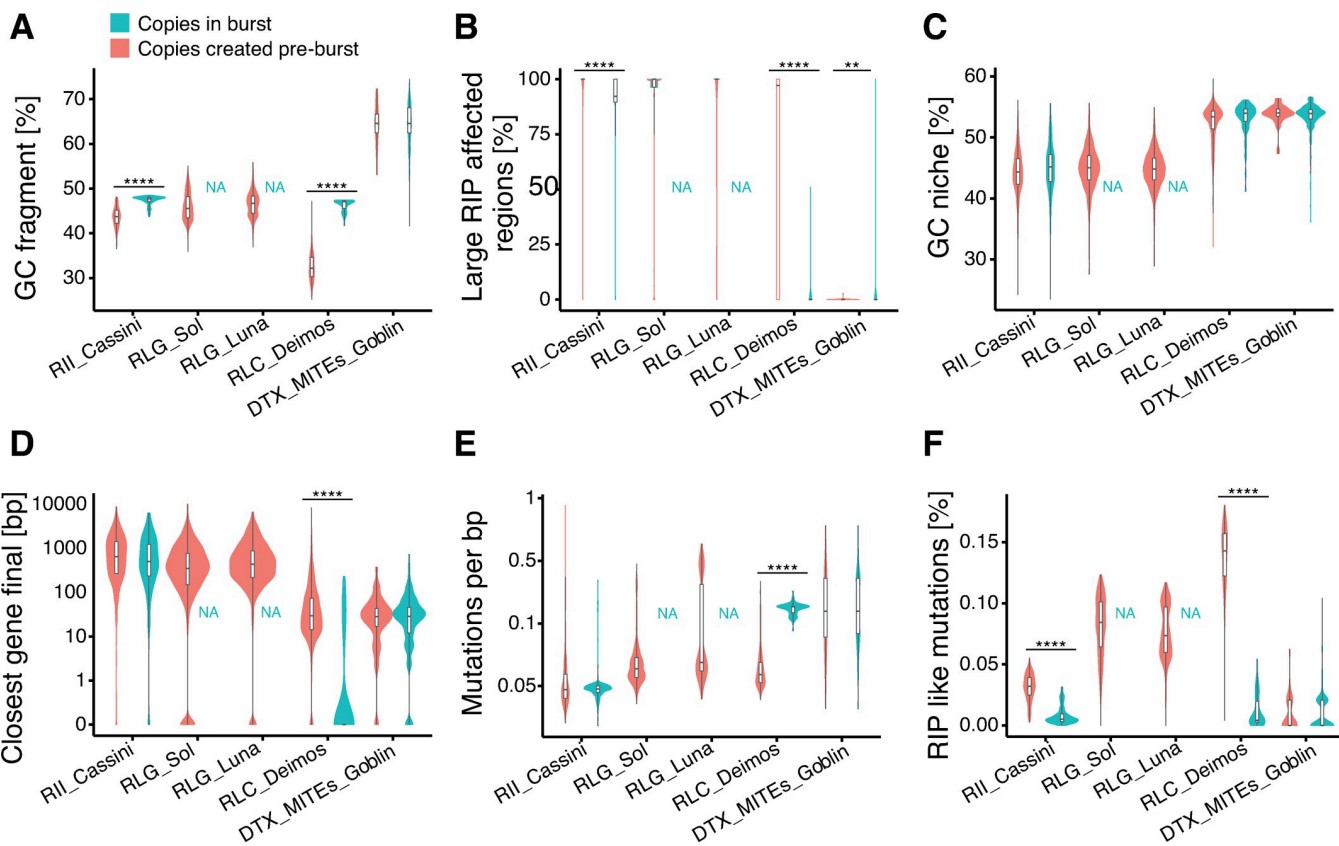

**Fig 6. High copy number TE families and characteristics of burst initiation:** Comparison of copies in bursts (red) and all other copies (blue) for the five TE families with the highest copy numbers. The TE families RLG_Luna and RLG_Sol have no copies assigned to burst clades (indicated by NA). The mutation rate was corrected for the length of the fragment and the number of copies across genomes.

Escape of RIP by RLC_Deimos is supported by the fact that the TE shows only single large burst following slow diversification (Fig 7A). The burst copies show clear changes both in niche characteristics as well as in sequence characteristics. The number of RIP-like mutations are gradually decreasing for copies closer to the burst, and new copies generated by the bust seem unaffected by RIP (Fig 7A). The TE niche retains a GC content close to the genome-wide GC content for both old and young copies (Fig 7B), yet TE copies in and close to the burst are almost exclusively inserted into niches devoid of RIP signatures (Fig 7C; see low LRAR). RLC_Deimos copies in general have a low GC content, yet the copies in the burst seem to have regained a higher GC content (Fig 7D). Finally, RLC_Deimos copies generated at the onset and during the burst have consistently been closer to genes compared to older copies pre-burst (Fig 7E). Interestingly, one of the putative copies leading to the burst of RLC_Deimos is found only in a single genome and inserted into a gene encoding an alpha/beta hydrolase.

Compared to RLC_Deimos, RII_Cassini has likely undergone six individual bursts of which two are very recent and specific to a single analyzed reference genome (from Australia and Canada, respectively). The tree structure shows two bigger clades, with one showing a generally decreased number of RIP-like mutations (Fig 8A). All of the bursts are located in a sub-clade with a decreased impact of RIP. Yet, the GC content of the niche is variable (Fig 8B), and almost all copies are located in regions with a high LRAR (Fig 8C). Copies of RII_Cassini generated during bursts have a higher GC content (Fig 8D). The change in GC content and the influence of RIP seems to be gradual, as most terminal branches have similar values compared

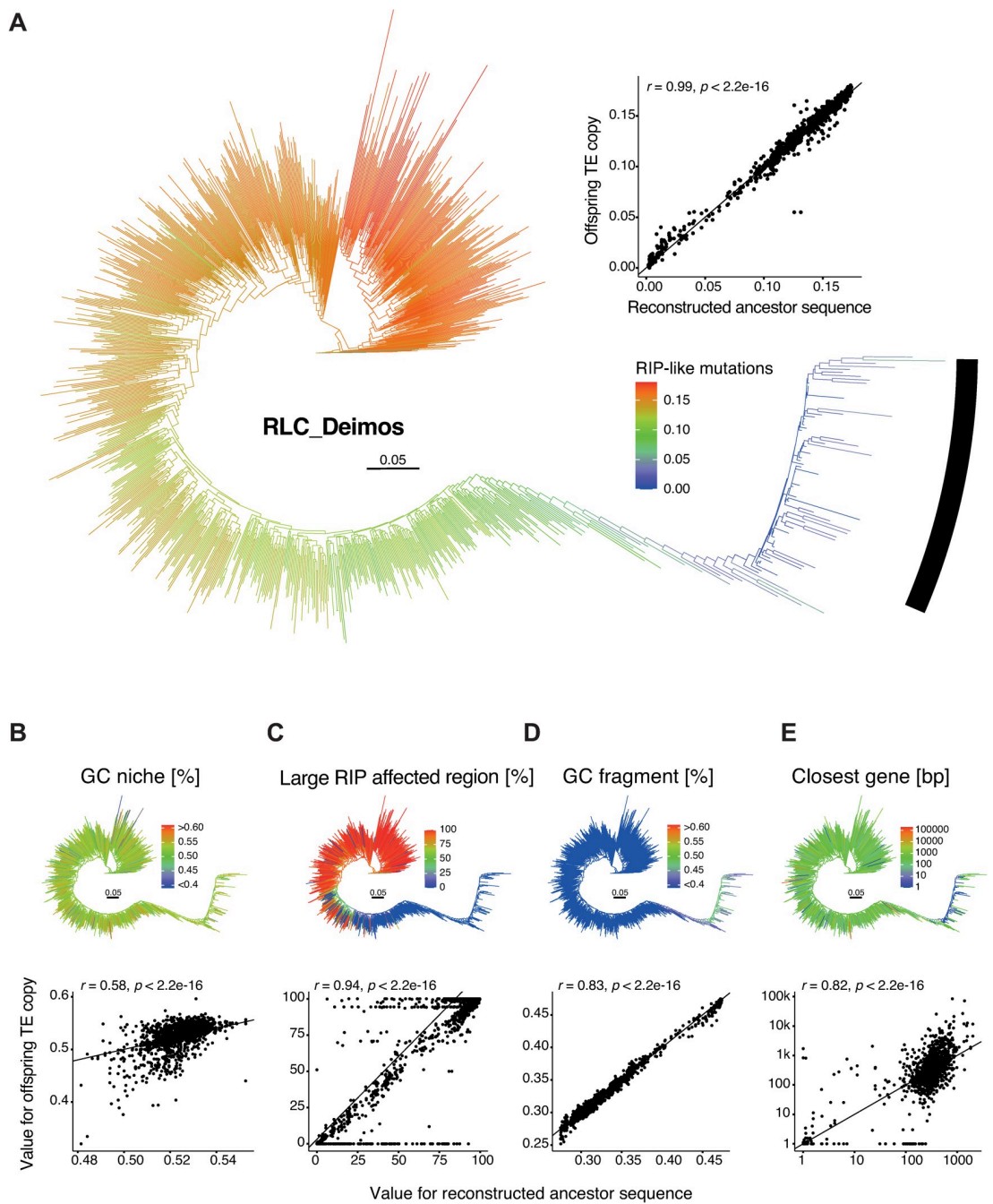

**Fig 7. Phylogenetic reconstruction of the *Deimos Copia* retrotransposon:** (A) Phylogenetic tree with colors indicating the number of RIP-like mutations. The black bar marks the different burst clades. The dot plot shows the changes in RIP-like mutations from the estimated ancestral state to the offspring for all internal and terminal branches from the ancestral state reconstruction. (B-E) Phylogenetic trees and ancestor-offspring changes for (B) the GC content of the niche, (C) the overlap of the niche with large RIP affected regions, (D) the GC content of the fragment and (E) the distance to the closest gene.

to the reconstructed ancestral state. Niche characteristics including GC content, large RIP affected regions or the distance to the closest gene are generally more weakly correlated between ancestor copies and derived copies compared to RLC_Deimos (Fig 8, dot plots).

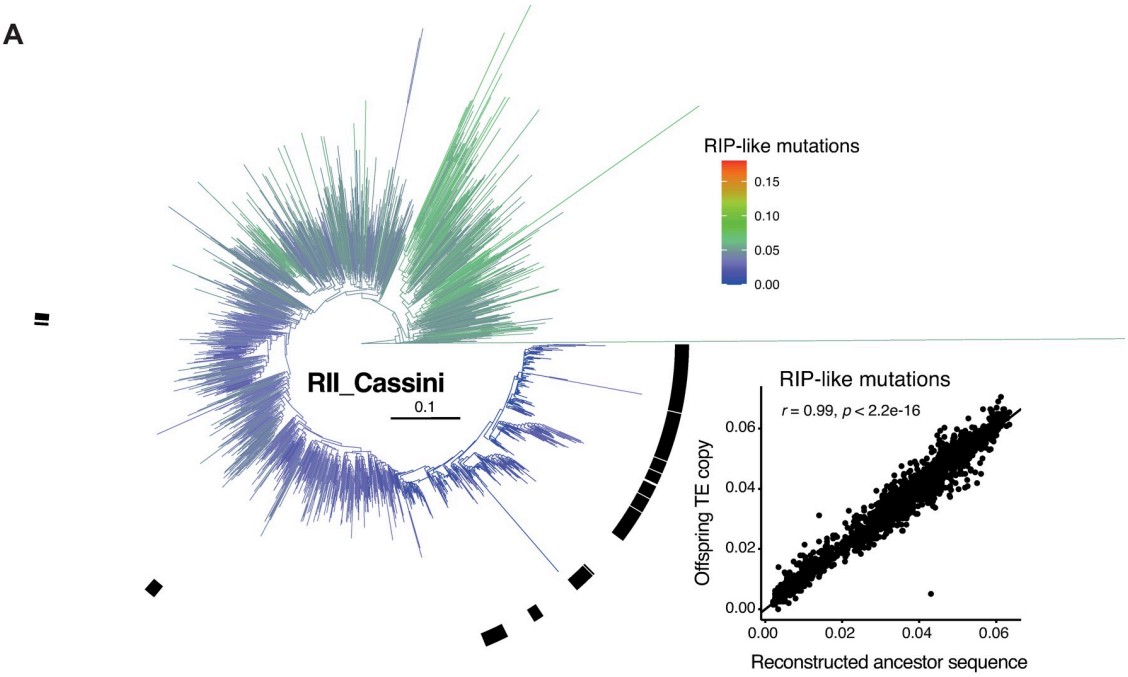

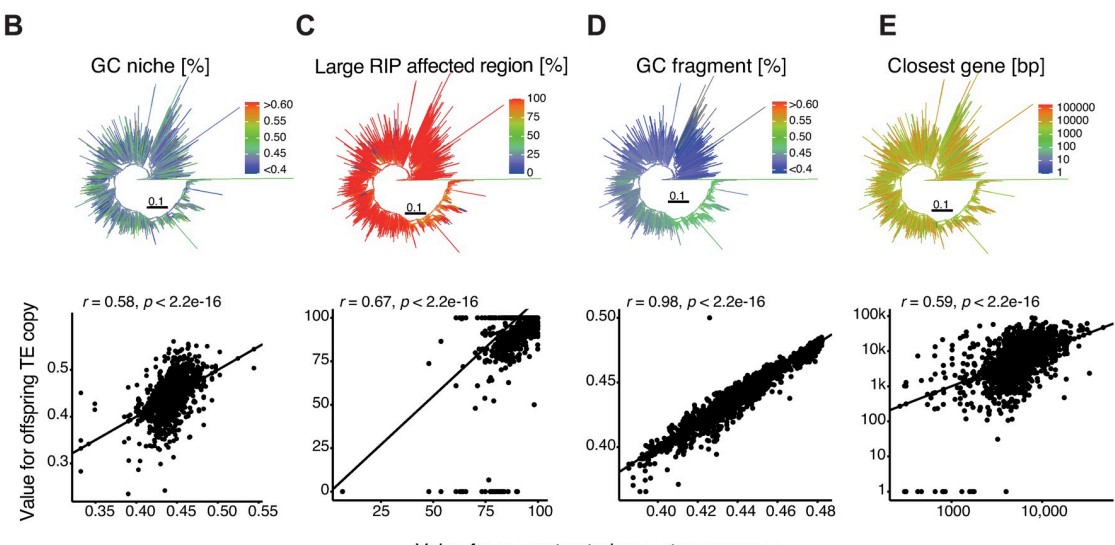

**Fig 8. Phylogenetic reconstruction of the *Cassini* retrotransposon:** (A) Phylogenetic tree with colors indicating the number of RIP-like mutations. The black bar marks the different burst clades. The dot plot shows the changes in RIP-like mutations from the estimated ancestral state to the offspring for all internal and terminal branches based on the ancestral state reconstruction. (B-E) Ancestor-offspring changes for (B) the GC content of the niche, (C) the overlap of the niche with large RIP affected regions (LRAR), (D) the GC content of the fragment and (E) the distance to the closest gene.

DTX_MITE_Goblin generally has high copy numbers among genomes with similar numbers and in orthologous positions, consistent with TE activity in the early history of the species rather than very recently. The expansion of the DTX_MITE_Goblin is characterized by a high number of bursts (S3 Fig). Copies from individual bursts are typically shared among all

genomes. Despite high nucleotide diversity, few mutations were generated by RIP. The DTX_MITE_Goblin family might consist of older TE copies shared between genomes, that are not affected by RIP. Taken together, high copy number TEs tend to have very recent burst origins, a higher GC content compared to the other copies and an ability to evade genomic defenses.

## Discussion

TE activity is an important disruptor of genomic integrity and the potential for deleterious effects is strongly influenced by selection and insertion site preferences. TE dynamics also play key roles in pathogenicity evolution by modulating effector functions and expression in fungal plant pathogens. Our joint analyses of 19 complete *Z. tritici* genomes demonstrate that multiple TE families have been highly active in the recent evolutionary history of this species. A substantial number of TEs have produced one or multiple bursts of proliferation that have distinct niche characteristics closer to genes and higher GC content compared with copies predating bursts. Recent bursts have mostly produced TE copies of the non-autonomous MITEs and the *Copia* family RLC_Deimos. MITEs likely escape the detection by genomic defenses including RIP due to their short length. However, the long copies of the RLC_Deimos were strongly affected by RIP prior to the bursts. The recent proliferation of RLC_Deimos copies without apparent accumulation of RIP-like mutations suggests reduced efficacy of RIP in general. Reduced RIP activity in some genomes is consistent with mutations and losses in the *dim2* gene implicated in RIP. Overall, genomic defense mechanisms were only partially effective against preventing the proliferation of TEs. Beyond the escape from RIP, our analyses indicate that a key trigger point for initiating new bursts is likely the successful insertion close to coding sequences. Hence, the genomic environment appears to play a key role in the evolutionary success of TE families.

We identified the emergence and expansion of numerous clades within TE families consistent with a burst of rapid copy number expansions. TE families often include large numbers of inactive copies that accumulate mutations and are unlikely to cause new insertions. Consistent with the dynamic nature of TE families, many TEs of *Z. tritici* are not detectable in the genomes of sister species. In particular, MITE families are most often absent in the sister species. This is consistent with rapid divergence of these non-autonomous elements devoid of coding sequences. Searching for more distant homologs to *Z. tritici* TEs might help identify additional putative ancestors, as observed in other species [45,68]. Rapid proliferation of new TE copies is highly uneven among TE families, with a small subset of families driving the majority of recent insertions. Such new insertions have persisted despite potential purifying selection against new insertions, silencing or mutations introduced by the genomics defenses (*i.e.*, RIP) shared by many ascomycete plant pathogens. TE families such as the RLC_Deimos were after an initial activity successfully repressed, but more recently reactivated starting a new branch or "subfamily" that originated from a particular subset of TE copies in the genome [49].

Active TEs that generate identical copies typically trigger defense mechanisms. Hence, such TE families are the most likely to be highly affected by RIP mutations. However, our results suggest RIP has only a weak impact on young TEs. Weak RIP signatures were predominantly found in MITEs, which supports the idea that smaller TEs most successfully evade recognition, as seen in *N. crassa* or *Parastagonospora nodorum* [19,69]. Longer elements show stronger impacts of RIP using both GC content and RIP-like mutations as proxies. We found evidence for RIP mostly in TE copies closer to the root, while copies generated during recent bursts show nearly no RIP signatures. Escaping the effects of RIP may be a prerequisite for the

initiation of a burst, hence the strong association of genetic distance and RIP mutations. RIP has mostly been studied in the Ascomycete *N. crassa*, where RIP introduces a large number of mutations after just one generation of sexual recombination [18]. The life cycle of *Z. tritici* is thought to consist of several cycles of asexual reproduction during the growing season, and only one round of sexual reproduction at the end of the season [70]. Hence, TEs may proliferate despite RIP for short periods but ubiquitous sexual reproduction should provide ample opportunity for RIP to act on copies of any recent burst. Evidence for RIP mutations are widespread in the genome of *Z. tritici* and the machinery for RIP is present in at least some isolates in the Middle East, the center of origin of the species [17]. Species-wide analyses have suggested that newly established populations of the pathogen have lost a functional RIP machinery [22,62,63]. Two isolate-specific bursts in the Canadian and Australian strain genomes for RII_-Cassini might indicate a spread of TEs after the loss of RIP. The geographic and temporal variation of RIP-mediated TE control in *Z. tritici* adds significant complexity to predict triggers of TE bursts. As the pathogen shows high degrees of gene flow even among continents, genotypes recovered from regions without active RIP may still present chromosomal segments with recent RIP activity due to admixture events introducing such RIP-affected regions. Hence, individual genomes are unlikely to carry homogeneous signatures of either RIP or consequences of TE activation due to a loss of RIP.

The only burst in RLC_Deimos is separating presumably older copies with very low GC content from burst copies driving the creation of a TE subfamily. The exact trigger of the RLC_Deimos expansion is unknown, however the ladder-like structure in the phylogeny predating the burst suggests a slow but regular pace of creating additional copies until the TE gained the ability to expand through a burst to create a large number of new copies. Loss of the RIP machinery would help preserve nearly identical copies of recently duplicated TE sequences and would maintain GC content at high levels. To what extent the loss of RIP has shaped the distribution of GC content across the genome including TEs copies remains unknown. The absence of RIP-like mutations in nearly all recently expanded TE families strongly suggests that recent TE proliferation was enabled by weakened genome defenses in the species. Yet, the absence of RIP cannot explain the strong increase in GC content in TE copies of recent burst, especially RLC_Deimos. The GC content is expected to remain low unless mechanisms such as GC-biased gene conversion increase GC content. Gene conversion is thought to have only a weak impact on the *Z. tritici* genome [71]

A mechanistic understanding of triggers activating TEs in fungal pathogens is largely lacking, yet here we show the importance of the niche harboring TE copies. Young TE copies have distinct associations with particular genomic niches compared with older copies, which tend to be located in niches with high TE content. In contrast, copies triggering recent bursts and the resulting burst copies themselves tend to be inserted closer to genes. However, the association of young TE copies with genomic niches is most likely confounded by the action of purifying selection. As TEs can disrupt coding sequences or change expression profiles of neighboring genes, purifying selection is most likely strongest against TEs inserting into gene rich niches. In contrast, TEs inserting into TE-rich niches possibly causing nested TE copies likely to have only a minor impact on fitness. In some fungal plant pathogens including *Z. tritici*, such nested insertions led to the compartmentalization of niches with high TE density and niches mostly composed of genes [72]. Repeated insertions of TEs into such TE-rich niches likely exacerbated genome compartmentalization. Selection is also expected to act on the effectiveness of defenses against TEs. Silencing or hypermutation of TEs close to genes may disrupt the functionality of the genes as well. Hence, the efficiency of genomic defenses against TEs may be weakened by selection. It is thus conceivable that otherwise silenced TEs can remain both functional and active when inserted close to a gene. As the TE copies created during the

RLC_Deimos burst are significantly closer to genes, this suggests that the proximity to genes provides a "secure niche", in which TEs will not be as efficiently targeted by defense mechanisms. RLC_Deimos copies predating the burst are generally in niches with a strong impact of RIP (*i.e.*, LRAR), yet all elements generated during the burst are in regions outside of LRAR. While LRAR are mostly made of TEs and other repeats, such regions are likely marked by heterochromatin and repressed. The insertion of burst TE copies into euchromatic regions close to genes may have facilitated to sustain TE activity. However, the benefits of inserting closer to genes to trigger bursts are not universal among TEs or insertion site preferences prevent switches from low GC to high GC content regions (*i.e.*, close to genes). This may be the case for RII_Cassini, which showed no clear association with distance to genes and burst triggers. Similarly, copies of DTX_MITEs_Goblin may have a universal preference of coding regions.

Beyond the benefit of weaker genomic defenses, the propensity of bursts being initiated by TEs inserting near genes may also be related to beneficial impacts of the TE copy itself. We found that copies at the start of bursts tend to have higher allele frequency within the species (*i.e.*, present in most analyzed genomes; see also S4 Fig). Larger pathogen population genomic datasets will enable the analysis of selection acting on parental copies initiating bursts. It is conceivable though that the beneficial effects of an individual TE copy are linked to triggers of TE bursts. However, selection at the organismal level driven by fitness benefits conferred by specific TE copies (*e.g.*, beneficial modifications of effector gene functions) is compounded by higher proliferation rates which benefits TEs themselves. Such multi-level selection was recently suggested to represent a *Devil's Bargain* in plant pathogens trading short term benefits of TE copies with longer term risks of genome expansions [35]. Assessing fitness effects of individual TE copies along with their expansion history will enable further hypothesis testing about the proximate drivers of TE expansions over a range of evolutionary time scales. Resolving drivers of TE dynamics will help predict the risk individual pathogens pose and how adaptive evolution is likely to proceed. This will require an integration of genome-wide TE dynamics with the consequences for host-pathogen interactions, and ultimately improve our mechanistic understanding of rapid evolutionary processes in plant pathogens.

## Methods

### Genome sequences and transposable element detection

We used a set of 19 reference-quality genomes of *Z. tritici* assembled using PacBio sequencing [52; European Nucleotide Archive BioProject PRJEB33986]. The genomes cover the global genetic diversity of the species with isolates originating from 14 countries and six continents (Fig 1A and S1 Table). We created the map using *ggmap* version 3.0.0 [73]. Map data originated from OpenStreetMap: https://www.openstreetmap.org/copyright. We used an improved TE annotation for the species with elements retrieved from all assembled genomes [52]. TE annotation steps included using RepeatMasker, LTR-Finder, MITE-Tracker, SINE-Finder, Sine-Scan and extensive manual curation with WICKERsoft and named based on the three-letter code [45,52,74–81]. The primary TE annotation was followed by stringent filtering steps to detect nested insertions and to join TE fragments. Simple repeats, low complexity regions and elements smaller than 100 bp were removed. TEs belonging to the same family overlapping by more than 100 bp were merged. TEs belonging to different families overlapping by more than 100 bp were considered as nested insertions. TEs belonging to the same family separated by less than 200 bp were considered as fragmented TEs and merged into a single element [52]. We additionally annotated TEs using the same pipeline in high quality genomes of the sister species *Z. ardabiliae*, *Z. brevis*, *Z. pseudotritici* and *Z. passerinii* [82].

## Multiple sequence alignments

We created multiple sequence alignments for all copies belonging to the same TE family from the 19 *Z. tritici* and four sister species genomes. We extracted all sequences of TE families with copy numbers ≥ 20 with the function *faidx* in samtools version 1.9 [83]. In case of fragmented elements, we extracted all fragments as individual copies. We reverse-complemented sequences where necessary prior to sequence alignment. To extract coding regions, we performed blastx searches against the PTREP18 database and against the non-redundant protein database from NCBI (09/2020) with diamond blast version 0.9.32.133 and selected the hit with the highest bit score with at least 200 bp length (Thomas Wicker; http://botserv2.uzh.ch/kelldata/trep-db/index.html) [84,85]. For small non-autonomous TE families lacking a coding region, we retained the entire sequence. We created multiple sequence alignments for each family with MAFFT version 7.453 and the following parameters: ––thread 1 ––reorder ––localpair ––maxiterate 1000 ––nomemsave ––leavegappyregion [86]. For four TE families with high copy numbers and large coding regions (RII_Cassini, RLG_Luna, RLG_Sol, RLC_Deimos), we slightly decreased accuracy of MAFFT, using the parameters ––6merpair instead of ––localpair.

## TE family divergence

TE families are expected to be active during different time spans and evolve at different rates. To estimate the genetic distance of the TE families, we ran RIPCAL with ––windowsize 1000 ––model consensus to create an additional consensus sequence that includes all copies of a TE family [87,88]. In R we created DNAbin objects with the R package ape version 5.3 and calculated nucleotide diversity of the multiple sequence alignments for each TE family with *nuc.div* in the package pegas version 0.13 [89–91]. To compare between TE families, we divided the nucleotide diversity by the length of the corresponding TE coding region. We estimated the genetic distance of TE bursts per family using RepeatMasker. To compare recent activity or bursts, we created a repeat landscape using *build Summary*, *calcDivergenceFromAlign* using Kimura divergence and *createRepeatLandscape* in RepeatMasker and visualized the results with ggplot [92].

## Genomic environment of TE copies

We described the genomic characteristics of niches containing TE copies. The distance between genes in the *Z. tritici* reference genome IPO323 is estimated to be around 2kb, the genome-wide GC is 51.7%, and the TE content is 19.1% [52]. Data on TE, genes, large RIP affected regions and histone mark distributions are shown available in S5 Fig. To scan the genomic environment including contain genes and TEs, we created 5kb windows up- and downstream of TE copies. Then, we calculated the TE and gene content based on TE and gene annotations, respectively, using the *intersect* command in bedtools version 2.28.0. We calculated GC content with the *geecee* tool in EMBOSS version 6.6.0 [93–95]. We also calculated the distance to the closest gene and TE with the *closest* command in bedtools. We used Occultercut version 1.1 with default parameters to detect isochores with low (≤ 49%) or moderate (> 49%) GC content [96]. We used The RIPper to identify large RIP affected regions in all analyzed genomes, and calculated the overlap of TE copies and RIP affected regions with bedtools *intersect* [97]. For the reference genome IPO323, we used available ChIP-seq information (http://ascobase.cgrb.oregonstate.edu/cgi-bin/gb2/gbrowse/ztitici_public/) to define the chromatin structure in niches around TEs [64]. To reduce effects of TE characteristics in downstream analyses, we grouped TEs into the major categories: MITEs, retrotransposons (*RLC and RLG*), LINE and others.

## Characteristics of TE copies

Many TE sequences in the genome are fragmented due to nested insertions or partial deletions. To improve the quality of multiple sequence alignments, we selected only TE coding regions for phylogenetic analyses (S6 Fig). We extracted the coding regions from multiple sequence alignments with *extractalign* from EMBOSS, based on the position of the coding sequences. We removed all sequences not covered by the coding region with trimAl -gt 0 version 1.4.rev15 (http://trimal.cgenomics.org) from the multiple sequence alignment, and removed fragments that contained more than 50% gap positions in the coding region [98]. For downstream phylogenetic analyses, we exclusively used the filtered multiple sequence alignment of the coding regions. We calculated the GC content of each TE coding region with *geecee* in EMBOSS. To quantify RIP-like mutation signatures, we extracted dinucleotide frequencies for each TE family alignment with *count* in the package seqinr [99]. To define locus specific TE dynamics, we identified first the closest up- and downstream fixed orthologous genes based on the annotation of the pangenome with *closest* in bedtools [52]. Next, we defined TE copies belonging to the same TE family and being located between the same fixed orthologous genes as orthologous TE groups. As most TE copies are singletons or only present in few isolates, we did not further investigate orthologous TE copies. Visualizations were made with ggplot [100].

## Maximum likelihood trees

We estimated maximum likelihood trees for all TE families with indications for recent activity and bursts in the species. We extracted conserved blocks of the coding region with Gblocks version 0.91b, using the following parameters: -t = d -b3 = 10 -b4 = 5 -b5 = a -b0 = 5 [101]. For each TE family, we included two sequences retrieved from the same TE in sister species genomes to root trees. We estimated maximum likelihood trees with RAxML version 8.2 [102]. For this, we generated 20 ML trees with each a different starting tree and extracted the starting tree with best likelihood with the following parameters: raxmlHPC-PTHREADS-SSE3 -T 4 -m GTRGAMMA -p 12345 -# 10 −−print-identical-sequences. We performed a bootstrap analysis to obtain branch support values with the following parameters: raxmlHPC-P-THREADS-SSE3 -T 4 -m GTRGAMMA -p 12345 -b 12345 -# 50 −−print-identical-sequences. Finally, we added bipartitions on the best ML tree with the following parameters: raxmlHPC-PTHREADS-SSE3 -T 4 -m GTRGAMMA -p 12345 -f b −−print-identical-sequences.

## Ancestral state reconstruction

We performed ancestral state reconstruction, using the characteristics of each TE family both on characteristics of the TE sequences or the niche of the TE copy as phenotype. We imported the best scoring ML trees created in RAxML into R using *read.tree* from the package treeio version 1.10.0 [103]. We rooted trees with *root* in the R package ape, using sister species sequences as outgroups. We converted tree objects to tibble objects with *as_tibble* from the package tibble version 3.0.1 in tidyverse version 1.3.0 and assigned each sequence in the tree metadata on TE characteristics using dplyr version 0.8.5 in tidyverse [104–106]. We then converted the tibble objects back to tree formats with *as.treedata* in treeio [105]. Using *fastAnc* and *contMap* from package phytools version 0.7–47, we performed ancestral state reconstruction for characteristics of the following continuous traits: gene density, TE density and GC content of 2.5 kb niches surrounding the TEs, closest gene, GC and RIP-like mutations per bp [107]. To estimate ancestral states for binary characteristics (GC rich *vs.* poor, core *vs.* accessory chromosome), we used *make.simmap* from the package phytools with an equal rates model and 100

simulations. We visualized the trees with *ggtree* version 2.0.1 [108]. To retrieve clades representing recent bursts, we created polytomy trees from binary trees, using the command *CollapseNode* from TreeTools version 1.4.4 in R at branch lengths smaller than 1.1e-05 [109]. For each burst clade, we defined the parental branch and an outgroup of the clade with *offspring* in the *treeio* package as the ancestral branch. The outgroup represents a copy outside but close to the burst clade. We compared niche metrics of ancestral branches of bursts with the distribution of the same metric in all elements outside of bursts. We performed associating mapping for shared characteristics along the phylogenetic tree using treeWAS [110].

## Supporting information

**S1 Fig. Hierarchy TE superfamilies: Classes, subclasses, orders, superfamilies as well as the tree-letter code according to Wicker et al (2007) [45].** The *Z. tritici* specific family names are according to Badet et al (2020) [52].
(PDF)

**S2 Fig. Characteristics of high-copy TE families: TE families are ordered from the highest copy numbers to lowest copy numbers (right) in all 19 analyzed genomes combined.** (A) Total copy numbers. (B) Long ($> 0.00001$; red) and short ($\leq 0.00001$; blue) terminal branch lengths of individual copies characterizing two classes of divergence times.
(PDF)

**S3 Fig. Phylogenetic tree of the TE family DTX_MITE_Gobblin: (A) Phylogenetic tree with colors indicating the number of RIP-like mutations.** The black bar marks the different burst clades. The dot plot shows the changes in RIP-like mutations from the ancestor to offspring for all internal and terminal branches from the ancestral state reconstruction. (B-E) Phylogenetic trees and ancestor-offspring changes for (B) the GC content of the niche, (C) the overlap of the niche with large RIP affected regions, (D) the GC content of the copy and (E) the distance to the closest gene.
(PDF)

**S4 Fig. TE copy frequency.** Comparison of TE copy frequency between outgroups of burst and all other TE copies.
(PDF)

**S5 Fig. Genomic environment of the *Z. tritici* reference genome IPO323.** Circos plot showing the genomic environment of the reference genome IPO323 (Dutch isolate). Description from outside to inside contains the GC content, gene content and TE content in windows of 10kb, the presence of large RIP affected regions (LRAR), and the indication of the histone marks H3K4, H3K27 and H3K9. Chromosomes 1–13 are core chromosomes that are present in each isolate, while chromosomes 14–21 are accessory chromosomes, that are not shared among all isolates.
(PDF)

**S6 Fig. Procedure to obtain multiple sequence alignments among copies of TE families.** Due to the high number of nested insertions and partially deleted fragments, we aligned only coding regions.
(PDF)

**S1 Table. Features of the global pangenome of *Zymoseptoria tritici* used for transposable element analyses.**
(XLSX)

**S2 Table. Description of all TE copies analyzed: assigned TE family, information about isolate of origin, position in the genome, niche and TE sequence characteristics.**
(XLSX)

## Acknowledgments

We are very grateful for helpful comments on an earlier version of the manuscript by Emile Gluck-Thaler, for discussions about phylogenetic inference with Vinciane Mossion and statistical advice by Claudia Sarai Reyes-Avila.

## Author Contributions

**Conceptualization:** Ursula Oggenfuss, Daniel Croll.

**Formal analysis:** Ursula Oggenfuss.

**Investigation:** Ursula Oggenfuss.

**Supervision:** Daniel Croll.

**Visualization:** Ursula Oggenfuss.

**Writing – original draft:** Ursula Oggenfuss, Daniel Croll.

**Writing – review & editing:** Daniel Croll.

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
