## [Decision Letter · Decision Letter 0]

26 Sep 2022

Dear Prof Croll,

Thank you very much for submitting your manuscript "Recent transposable element bursts triggered by insertions near genes in a fungal plant pathogen" for consideration at PLOS Pathogens. As with all papers reviewed by the journal, your manuscript was reviewed by members of the editorial board and by several independent reviewers. First of all, I would like to apologise for the lengthy review process. However, I had difficulties finding appropriate reviewers at the start, and later on right after, the holiday season. But eventually even four reviewers agreed that are quite unanimous in their assessment. In light of the reviews (below this email), we would like to invite the resubmission of a significantly-revised version that takes into account the reviewers' comments.

We cannot make any decision about publication until we have seen the revised manuscript and your response to the reviewers' comments. Your revised manuscript is also likely to be sent to reviewers for further evaluation.

Sincerely,

Bart Thomma

Section Editor

PLOS Pathogens

Bart Thomma

Section Editor

PLOS Pathogens

Kasturi Haldar

Editor-in-Chief

PLOS Pathogens

orcid.org/0000-0001-5065-158X

Michael Malim

Editor-in-Chief

PLOS Pathogens

orcid.org/0000-0002-7699-2064

Reviewer's Responses to Questions

**Part I - Summary**

Reviewer #1: In many plant pathogenic fungi as well as other organisms, transposable elements (TEs) exist abundant in genomes and contribute to their evolution. In these fungi, however, little is known about how bursts of TE copies have occurred, due to the insufficiency of high-quality genome assemblies. This paper characterized TE burst events of a wheat pathogen Zymoseptoria tritici using complete (telomere-to-telomere) genome sequences of 19 isolates. By a phylogenetic approach, the authors divided TEs with higher copy numbers into the copies derived from burst events and the others. The analysis revealed that certain TE families experienced the bursts one or several times and that TEs in the burst clade tended to be located closer to genes. This paper showed the association between the TE bursts and the genomic environments at each TE family level. This paper along with the other accumulated evolutionary papers of this fungus can contribute to the understanding of how the pathogen evolution has been driven, which can give a certain impact on related research fields of other plant pathogens. However, there are fatal data labeling errors and untransparent descriptions, which make it harder to judge whether some of the authors’ interpretations are correct or not, as listed below. These data should be reanalyzed and reinterpreted carefully. Even though the mistakes, the overall approaches toward the scientific question seem reasonable so the manuscript will be fine as long as the errored data and accompanied interpretations will be corrected without any contradiction.

Reviewer #2: I think that the paper summarize a good work.

The authors uses 19 isolates to describe the history of TE evolution in different Z. tritici. They characterize all TEs in all 19 isolates, they find specific characteristics of each TE family and using two of them, they try to describe burst of expansion. It is known that genome evolution is shaped by TE burst and therefore, they try to connect TE burst to evolution of individual strains isolated in different part of the world.

although the paper is readable, in some parts there are many assumptions and in the last paragraph of the results, there is little indication where to find the interesting results in the figures. Ultimately, the paper results difficult to read for people that are not in the field of TE evolution

Reviewer #3: The manuscript “Recent transposable element bursts triggered by insertions near genes in a fungal plant pathogen” by Oggenfuss and Croll is a fascinating deep dive into the transposable element dynamics of the fungal pathogen Zymoseptoria tritici. Taking advantage of a recent population genomic dataset of 19 strains, the authors take a unique approach of evaluating the genomic niche that different TEs occupy within the genomes. They determine what properties of these niches correspond with given TE superfamilies and uncover interesting patterns with regard to which elements are currently undergoing expansions within the population. The work is very methodical and encompassing, however I feel that a lot of improvement should be made before a final version is ready for a broader scientific audience. In particular, I believe that the current draft will be very difficult to understand for readers that are not experts on transposable elements. I hope that the following comments will help the authors rework their manuscript.

Reviewer #4: I have now read and considered the manuscript entitled Recent transposable element bursts triggered near genes in a fungal plant pathogen

Overall this is a well written manuscript that leverages 19-complete telomere to telomere assemblies to examine the evolution of transposon families that have undergone recent ‘bursts’ within Zymoseptoria tritici. The authors have uncovered some interesting patterns, for example that small non-autonomous transposons (mites) are more likely to escape genome defence mechanisms and reside near genes. The authors also explore in more detail two very large transposons that have undergone recent expansion (Cassini, Deimos). I do not find any flaws in the analyses or data presented, however I found the amount of information presented in the figures overwhelming, which made it difficult to distil out the main messages that the authors wish to communicate. I realise that this is likely a shortened revision of a previous manuscript based on my observation that the text references 5 figures but there are seven figures in the material. Also, there are more panels listed in the figure legends than there are in the figures. So I suspect that I am asking the authors to even further simplify a longer version of a prior draft manuscript.

Below are some comments/suggestions that I hope will improve clarity and help the authors better communicate their main findings.

**Part II – Major Issues: Key Experiments Required for Acceptance**

Reviewer #1: 1. There are several fatal errors in Figure 3.

1.1. Some TE families seem mislabeled in Figure 3A-E, which leads to the contradiction with Figure 3F and texts. For example, the description in lines 177-178 contradicts Figure 3B. The median for DTX_MITE_Goblin and RLC_Deimos (the fourth or the fifth from the left) seems below 40% or over 60%, respectively, in the figure, against the text description. Moreover, the values shown in Figure 3B-E do not correspond to that shown in Figure 3F. It can be easily understood from the plots of DTX_MITE_Goblin [the highest RIP-like mutation in (E)], RLC_Deimos [shorter length in (C)], and RLG_Uranus [the highest nucleotide diversity in (D)], for instance.

1.2. It seems strange that the scale for “RIP-like mutation” in Figure 3E and F is inconsistent, without any explanation.

1.3. The explanation for each panel in the legend is off by one or two, probably because two panels [former (A) and (G)] were moved to Supplementary Figure S2A. Please correct the alphabet number in the Figure 3 legend.

There are additional suggestions for Figure 3 as follows.

1.4. Please add the explanation to the legend for what the red horizontal line in Figure 3B and C indicates. In line 176, please describe what percentage is “the genome-wide GC content”. Does the red horizontal line in Figure 3B indicate that?

1.5. In Figure 3D, can the nucleotide diversity be shown by a logarithmic scale? Otherwise, it is hard to understand that “MITEs tend to have higher nucleotide diversity” as described in line 184.

1.6. If the box plots and/or bar graphs in Figure 3A-E will be colored distinguishable for each superfamily, it will be easier to catch some trends discussed.

2. There are also a suggestion and errors in Figure 4.

2.1. Why DTX_MITE_Goblin was excluded from the analysis in Figure 4A?

2.2. In Figure 4D, what the colors indicate is inconsistent between the illustrated or text legends. The red and blue colors respectively show “no burst” or “burst” on the figure, but opposite in the text legend in lines 760-761.

2.3. The discussion in line 261 is inconsistent with the result in Figure 4D. The “lower” would be “higher”.

3. Some results are described without showing data, exemplified as follows. It is desirable to visualize or summarize the data in additional figures or tables.

3.1. The description in line 155, “TE insertions are at higher density on accessory chromosomes (Figure 2B)”, is not supported by Figure 2B which shows “number of copies” but not “density”.

3.2. As for the description in lines 217-218 (“Overall, around 50% ... several bursts”), could it be shown which TE families experienced how many times bursts in how many genomes out of 19?

3.3. Could the data for a discussion in lines 322-324 (“We found ... most analyzed genomes).”) be shown?

4. Please add a clear explanation for the difference between “RIP index” and “RIP-like mutations” analyzed in Figures 2F, 4B and D.

5. The trend that burst TE copies tended to be located near genes (Figure 4D) is not observed in some TE families (e.g. RII_Cassini and DTX_MITEs_Goblin) (Figure 5). Please add the discussion about that.

6. It seems better to change the paper title from “… triggered by …” to “… associated with …” or other such milder expression, since this paper shows no direct evidence for the causal relationship expressed by the current title. Lines 130-131 (“The comprehensive … number expansions.”) and 263-264 (“Our analyses ... coding sequences.”) also seem to be over-discussed.

Reviewer #2: Line 140: positive correlation needs to be assessed with statistical means.

Line 147: what do you mean with the sentence “Only few TE insertions are fixed among genomes (n = 122; Figure 1D) and consist predominantly of MITEs”. Reading the paragraph, I understand that it refers to a specific family present in multiple isolate or only in one but reading the figure legend is more about location rather than presence/absence of a family:” Allele frequencies of TEs at orthologous insertion loci among genomes”. Can you explain better in the text?

Line 150: I understand what the author did but I think that it should be explained better. I can understand that the author think that all readers need to know which marker are associated with euchromatin and which with heterochromatin but at the moment I think that this is not a common knowledge therefore I think that you need to explain which marker goes with either heterochromatin or euchromatin

Line 155: “, but TE insertions are at higher density on accessory chromosomes”. Honestly, I can not see where this is reported in fig 2B. In contrast I see that that bar in figure 2B regarding the panel chromosome has a higher number of TE in the core genome. I think that the authors should find a different way to plot the density in function of core/accessory chromosome size

Line 184: “MITEs tend to have higher nucleotide diversity at 185 similar copy numbers compared to other TE families (Figure 3D, F)”. I find difficult to see this. In figure 3D there are many MITs elements but only one has an high Nucleotide diversity: DTX_MITE_Toll. The rest is normal.

Line 185: looking at RIP, on MITE_Goblin has an high value of RIP compared to others.

Line 193: “Our findings show that recent bursts are characterized by high copy numbers, but genomic defences and the length of TEs create complex outcomes for individual TE families.” Honestly, I do not understand where this come from and what this means

The paragraph entitled “NICHE CHARACTERISTICS OF TRANSPOSABLE ELEMENT ACTIVATION” is very difficult to read. The paragraph is short and refers to fig 5 and 6 where for each figure A-E panels. The authors describe their findings but do not refer in an appropriate way where to find the information that support their statements. This paragraph should be written better and explained better the results

Reviewer #3: I have only one major concern with the analyses, which involves RIP. Given that RIP is a stochastic process, some of the underlying assumptions that the authors appear to have made may be violated. Firstly, RIP becomes inefficient for repeat lengths below ~ 500 bp (see https://doi.org/10.1038/ncomms4509), thus the smaller TEs like MITES are likely undergoing fundamentally different evolutionary processes. This could cause a number of issues when lumping all of the analyses together. For example, using the same branch length cut offs for both the MITEs and LTRs could be underestimating the number and nature of burst events. Second, it seems that the authors have come to some concrete conclusions about precisely which element is the ‘master’ copy of burst events. It is not clear from the current draft exactly how this was done, but both the nature of RIP, and the fact that the MITEs are non-autonomous, should make this very difficult if not impossible. Lastly, as the authors bring up in the discussion, some strains have lost RIP or components of it, which would mean that any TE dynamics in those strains would be fundamentally different from the others. It is not mentioned which strains have lost it, and as the results never break down the dynamics by strain, one cannot currently evaluate this. Some of these points could directly impact the main conclusion, that bursts are driven by insertions near genes. For example, is the reported copia element burst in a strain without RIP? Is it then true that bursts originate from gene rich regions, or does that strain have a unique distribution of TEs because it has lost its genome defense machinery? Likewise, although the MITEs are in gene rich regions, where are the protein coding elements? Isn’t that the more important question with regard to what’s driving the expansions?

In addition to this issue, the introduction could be improved. Large parts currently read as a list of interesting facts that are only loosely tied together, and are sometimes repetitive. e.g. Line 54 talks about silencing of TEs, and the same information is presented again on Line 120. Other important information is either hidden in the supplement (like what the codes RLC, RLG etc. mean, or even definitions for Copia and Gypsy), or not presented until the discussion, such as much of the information about RIP. Furthermore, the start of the results section seems to confound analyses from the previous study (Badet et al., 2020) with what is done here. e.g. Line 140: "show a positive correlation", this does not appear evident from Figure 1, but is clear from Badet et al., 2020 figure 5A. The authors should be clear about what information has been presented again for the sake of this study and what are new analyses.

Reviewer #4: The authors should include a better introduction to transposon families, in particular the three families that they focus on throughout the results (MITEs, Copia, Gypsy).

Introduction: For non-specialist readers it may be helpful to briefly introduce Class I vs. Class II TEs. Many of the TE names listed in the Manuscript use the Wicker classification abbreviation (RLX, RLC,RLG) but this is never defined. May be enough to define Copia/Gypsy, LINEs and MITEs? I also note a new name for the TE family Gypsy has been proposed, to avoid some of the negative history associated with this name, I wonder if the authors would consider adopting this new name. Lines 96-101 may be a good place to revise and consider adding this information in, the current text here describes the loss of Long-terminal repeats will probably not make much sense to someone who is not familiar with TE families that contain these features.

The loss of dim2 in some populations of Z. tritici is an important point that is not very clearly linked to the results where two individual genomes contain significant TE bursts. Is the Australian isolate one of these genomes? More clearly signposting this in the results (or even in Fig1?) will improve clarity as to why TE bursts are more likely to be observed in some isolates.

I have struggled with the ancestral reconstruction analysis and phylogenetic trees in the context of RIP. It was not clear to me from the methods if the AR was conducted on “de-RIPed” alignments? In my experience de-RIPing or removing sites in an alignment with RIP-like mutations increases pairwise sequence identity of alignments dramatically. I have concerns that the clades with high RIP mutations in figures 6-7 are grouping mainly on AT richness (text figure 5?), would trees constructed with de-ripped sequences give a better representation of the relationships within this TE family? I can see clearly for Deimos that the burst clade has very low RIP. This is less obvious for Cassini because the bursts are much smaller and a little nested within the tree. I did not understand the dot plots in all parts of Figure 6-7 and the statistical tests associated with these. More clarity around this analysis may address this question I have about ancestor reconstruction and the link to RIP-like mutations but these figures are not discussed in the current text.

All figures: Please consider reducing the number of panels. What are your main messages, can you remove “GC fragment” or “Large RIP affected Regions” etc etc? Do these different categories show very different things? If yes, then this should be more clearly discussed in the text for each figure, where this data remains.

Figure 1: Parts A and B These are re-purposed figures from previous work, do you need to re-present both here, or is part B sufficient to show genome size is correlated with transposon copy number? Is the date of sampling important for isolates with TE bursts? Or perhaps indicating which isolates have a functional dim2 gene and therefore are capable of RIP?

Fig1D: Allele frequencies? Allele counts? Frequency is between 0-1? May consider moving text “insertions were considered orthologous…”to main results text rather than the legend

Figure 2: Please consider moving the Legend or making it more visible. Currently it is embedded in the middle of part C and not clear that it is the color legend for other parts of the figure too.

Figure 4D, Figure 5: are all these panels needed?

Figures 6-7: What is part D “GC content of the copy”?

Could the authors please provide the fasta files used to construct phylogenetic trees? In their public datasets on github (if not too large) or Zenodo database?

**Part III – Minor Issues: Editorial and Data Presentation Modifications**

Reviewer #1: INTRODUCTION

Line 54: Please remove “and chromatin”, redundant with “histone modifications” in lines 53-54.

Line 110: Better to add “(Mycosphaerella graminicola)” after “Zymoseptoria tritici”.

Line 119: Please explain “MgDNMT” briefly (e.g. a DNA methyltransferase).

Line 131: “Escape of” → “Escape from”

RESULTS

Line 151: “5kb” → “5 kb”

Line 159: Please mention which TE category in Figure 2 (“MITE”, “Copia/Gypsy”, “LINE” or “other”) corresponds to “RLC” and “RLG”.

Lines 212-213: The percentage “20.7%” is not “more than half”.

Lines 236: The adverb “Similarly” is confusing since the traits mentioned in this sentence (LRAR and closest gene of burst copies) seem opposite between RII_Cassini and RLC_Deimos.

DISCUSSION

Line 335: “our a” → “our” or “a”

METHODS

Line 362: Delete “(Supplementary Figure S4)”.

Lines 396: “TheRIPper” → “The RIPper”

Line 405: Insert “(Supplementary Figure S4)” instead of it deleted from line 362.

REFERENCES

Lines 488-489: Please add the journal and volume.

Line 547: zymoseptoria → Zymoseptoria

Lines 582-584: The paper is now published (doi: 10.1093/g3journal/jkab068). Please cite the peer-reviewed one.

Lines 601-603: Please add the journal and volume.

Lines 654 and 655: Please cite the URL.

FIGURES

Figure 1D: Why is the font size of “9” on the x-axis larger?

Figure 2A: Please describe what the x-axis represents.

Figure 2B legend: Please add “(LRAR)” after “large RIP affected regions”.

Figure 2F: Could the statistical significance be shown? (e.g. Put an asterisk to the correlation coefficients supported with p < 0.05)

Figure 5: Please explain “Mutations per bp” in the legend briefly. Is the mutation rate based on the consensus sequence for each family?

Supplementary Figures

Figure S3 title: “DTX_MITE_Gobblin” → “DTX_MITE_Goblin”

Reviewer #2: The authors should point better to the panels in each figure to help the reader to follow what they claim

Reviewer #3: Line 35: Is this only purifying selection or could there be other insertion preferences etc?

Line 43: a genomic environments or genomic environment.

Line 49: define abbreviation (TEs)

Line 51: "the potential of deleterious insertions into coding and regulatory regions", confusing wording, consider rephrasing.

Line 59: "in all copies". RIP can target all copies, but does not necessarily hit every copy due to its stochastic nature.

Line 85: explain positive effect.

Line 96: RIP will also create long branches

Line 111: co-evolved in what way? Is it an obligate pathogen in wheat? Is it specific to cultivated varieties?

Line 113: comma after reference

Line 141: In Supplementary table 1 there are several isolates with a lower TE-content than 18.1%.

Figure 1 seems to reproduce some information from the Badet et al., 2020 paper, such as the map. Is this necessary?

Figure 1A: Difficult to see differences in TE content in figure 1A, scale only goes between 17-24 (very small). Australia has no estimated TE-content, or is it the most?

Figure 1B: Figure text on x-axis is tiny.

Figure: 1C: What about non-MITE DNA transposons, are they all “other”? Both figure and text make it unclear what the abundance of these elements are. With this number of MITEs there must be some master copies, which are hijacked by the MITEs. They are likely <20 copies.

Figure 1C: “Between the same set of orthologous genes”, what if the genes are on the border of a large TE-island? Is this comparable to small insertion sites?

Figure 1 C, a lot of windows are 100% TE. Presumably this is due to nesting. Does that affect the results?

Table S2 contains a massive amount of information. This is good in terms of reproducibility and open science. However it makes it difficult to find the information pointed to as in text references. e.g. Line 145. Perhaps a more readable table with a summary of important information, like what superfamilies given families are in, is needed.

Line 145-146: Neither figure 1C nor supplementary figure 1 are showing good comparisons of the number of families belonging to superfamilies in the different genomes.

Line 151: why 5 kb? Does “centered around” include the TE itself, or only the flanks? If the former then many of the retrotransposons may be larger than 10 kb and would have skewed values due to this.

Line 156 and 159: Is the GC association only due to RIP?

Supplementary figure 1: All text is way to small

Figure 2A: Label X axis. Y axis is TE density? The colours should be stated here as well, this is an issue for many of the figures. I recommend either clearly having the legend in part A, or prominently at the side in each figure.

Figure 2B: The fact that there are more TEs in core likely reflects that there is more DNA in the core. Should show as a proportion. Same with telomere and LRAR.

I question whether setting a cutoff for low GC at < 50% is meaningful when the genomes contain 51% - 52% GC.

Figure 2 C: What are the weird peaks around 43% and 49%?

Line 168: "Longer TE copies tend to be located in already TE rich niches", is that length fragment? Throughout the text and figure 2 it is not clear what “fragment” refers to.

Supplementary figure 3A: are all clades bursts? The black bar on the outside is not that informative. The axes should be labeled clearer, not evident what's meant by copy and sequence.

Line 175: “We find that high-copy TE families tend to also have more variable copy numbers among 19 genomes, indicating ongoing activity of individual families”. Why does this indicate ongoing activity? Also, this is more of a discussion point.

Line 179: “with the shorter copies belonging to the non-autonomous MITEs” And what is the longest copy?

Line 182: RIP is not clock-like, so it cannot be considered “age” for ripped sequences.

Figure 3: I think the figure has labeling issues. It looks like the fourth column is DTX_MITE_Goblin, which has a consensus length of 226 bp, but the bar plot seems to show that it's over 5000 bp. Likewise the fifth column should be RLC_Deimos at 6356 bp, but the plot shows a very small size, currently unreadable. I didn't go through every column, but this should be checked.

Figure 3C: log scale would be better to show lengths as many look like 0. Perhaps for nucleotide diversity as well.

Figure 3F: The text in the plot is too small. It is also difficult to tell what TE Superfamilies the different data points correspond to (except for MITE).

Figure 3: There is no panel G or H? These sound like the same as supplementary figure S2A and S2B

Line 198: 'the factors'

Figure 4 B: I don't understand this plot at all

Line 200: What superfamilies do these belong to? It is quite difficult trying to figure out what type of elements these are.

Figure 5: The colours do not show up on many of the violin plots. Another way of labeling would help.

Figure 5: Colors in legend and figure text do not match. Legend: “Red not part of burst”. Text: “copies in burst (red)”

Figure 5: Confusing to have families that have both burst/non-burst and families that only have non-burst(?) in the same figure.

Figure 6: It looks like there may be a lot of bursts of intermediate age, does this affect the results? What do the dot plots show? It’s not clear what the x axis refers to.

Figure 6 - 7: Do trees have support?

Line 260: 'of TEs'

Line 384: While Kimura distances can be calculated on RIPped sequences, they mean something very different from how Kimura distance is normally discussed in TE literature i.e. these are not mutating at constant rates and don’t refer to age.

Reviewer #4: Minor comments:

Lines 78-79: Preferred insertion site of TEs, is it worth mentioning here that many of these are “TA” sequences, which is also linked with RIP?

Lines 151-152: is TE “insertion” misleading? You analysed 5kb windows around each annotated TE? Or you only analysed TE’s where you clearly had an empty site in another genome? (possible as very few TEs fixed)

Line 173-174: RLC and RLG not defined, Focus of figures 1 and 2 are mainly of these families and MITEs, might be worth clarifying early in results why you focus on these families and define them.

PLOS authors have the option to publish the peer review history of their article (what does this mean?). If published, this will include your full peer review and any attached files.

Reviewer #1: No

Reviewer #2: No

Reviewer #3: **Yes: **Aaron Vogan

Reviewer #4: No
---

## [Decision Letter · Decision Letter 1]

13 Jan 2023

Dear Prof Croll,

Thank you very much for submitting your manuscript "Recent transposable element bursts are associated with the proximity to genes in a fungal plant pathogen" for consideration at PLOS Pathogens. As with all papers reviewed by the journal, your manuscript was reviewed by members of the editorial board and by several independent reviewers. The reviewers appreciated the attention to an important topic. Based on the reviews, we are likely to accept this manuscript for publication, providing that you modify the manuscript according to the review recommendations.

Sincerely,

Bart Thomma

Section Editor

PLOS Pathogens

Kasturi Haldar

Editor-in-Chief

PLOS Pathogens

orcid.org/0000-0001-5065-158X

Michael Malim

Editor-in-Chief

PLOS Pathogens

orcid.org/0000-0002-7699-2064

Reviewer Comments (if any, and for reference):

Reviewer's Responses to Questions

**Part I - Summary**

Reviewer #1: The manuscript has been improved after the first review process. The discussion becomes deeper, which attracts interest in the relationship of TE bursts with the insertion into specific gene niches and the activity of a genomic defense mechanism (RIP). The finding of the variation of possible burst mechanisms among TE families would be also interesting. The paper can provide a subset of arranged information for TE science in plant pathogens. I suggest modifying some minor issues listed follows.

Reviewer #3: The revised version of the manuscript is a significant improvement on the original. The authors have done an excellent job addressing all of the reviewers comments. The clarity has increased dramatically, making for a pleasurable read. I think the manuscript is more or less ready for final publication in its current form, though I have noted some minor errors and have a couple suggestions listed below. However there is one point that I highly recommend the authors look over closely. In Figure 6 D it appears the burst copies of RLC_Deimos are mostly inserted into genes. I would double check these to make sure that the TE ORFs aren't miscalled as genes. If it's not an error this fact could be worth highlighting more in the text.

Additionally, I am aware of the debate regarding the family name "gypsy". I highly recommend using Ty3 as the name for the family, or at the very least Ty3/mdg4, especially as these are fungal TEs. In any case please check the consistency as sometimes RLG is used and sometimes mdg4 through out the manuscript.

Reviewer #4: (No Response)

**Part II – Major Issues: Key Experiments Required for Acceptance**

Reviewer #1: The major issues pointed at the first review process seems to be solved and I did not find additional major issues.

Reviewer #3: (No Response)

Reviewer #4: (No Response)

**Part III – Minor Issues: Editorial and Data Presentation Modifications**

Reviewer #1: Line 142: Delete "The loss of".

Line 211: RLC_Deimos seems an extreme case of "The GC content of high copy TE families" "lower than 50%" rather than "the exception".

Line 225: There is no "Figure 4H".

Line 246: The percentage of "a large part of TE copies" that "remain in large RIP-affected regions" is missing.

Lines 268-270: The description about RIP seems redundant within the sentence.

Lines 275-276: RII_Cassini seems to show a "similar pattern of temporal escape

from RIP facilitating a burst" like RLC_Deimos though with a smaller degree.

Line 663: Correct "xx".

Line 709: 67-U139 → 67-77

line 711: Bmc → BMC

Fig 1B: Why Copia/mdg4 instead of RLC/RLG?

Fig 7E, 8E and S3E: The values of the scale seems inconsistent between the pylogenetic tree and the correration plot. If the scale for the correlation plot indicates log10 value, please mention that.

Reviewer #3: Line 120: "high number of low highly similar" -> confusing language, consider rephrasing.

Line 137 : "Has both produced both" -> remove one both; missing an "and"

Line 189: "is" -> are

Figure 1: details of map construction should be in methods

Figure 2: the LRAR density pattern is surprising, is it labelled properly?

Line 197: "LINES" no s

Figure 4: Much improved, great work!

Line 303: "General" -> generally

Line 316: "play also" -> also play

Line 402: One reason why copies near genes may not be targeted is due to RIP spill over, where adjacent sequence is mutated along with the targeted duplicate region. This could be mentioned here to explain the "secure niche".

Supplemental figure 5: "Netherland" -> Dutch; error with "accessory chromosome" labels.

Supplemental table 1: are the underscores in the "isolate" column intentional?

Reviewer #4: This is a substantially revised manuscript that addresses most of the issues raised in my previous review.

One outstanding area that I feel could be explained more clearly is the ancestral reconstruction dotplots shown in figures 7 and 8. I still struggle to fully understand this analysis and feel that some additional explanatory text in the results here may help clarify, or in the methods lines 523-529. For example the phrase “We imported the trees into R using read.tree..” Which tree did you import as the starting point for this analysis the highest scoring ML tree? OR “We added metadata using ddplyr…” Which I assume means you assigned each sequence in your tree (each tip) the data associated with that particular TE copy. Would the authors like to state more clearly what the significance of a strong correlation or lack there of means? (i.e. strong correlation indicates not a lot of change through time? Or the inverse?)

Other minor typos spotted throughout the text:

Line 71: tough � though

Line 142: “the loss of cytosine methylation was lost..”

Line 160: “Escape from genomic defenses including silencing is a likely the major driver of TE dynamics.” This sentence stuck out to me because silencing was not really measured in this study.

Lines 174-176: “Half (n=106) … half to retrotransposons (n-59)”. Does not seem quite correct to use “half”.

Line 189-190: “We found no overall association between TE copies and large RIP-affected regions.” I think this comes with a small caveat that LRAR can completely destroy some TE copies making them impossible to detect.

Line 324: insert “by” …were strongly affected BY RIP prior to the bursts.

PLOS authors have the option to publish the peer review history of their article (what does this mean?). If published, this will include your full peer review and any attached files.

Reviewer #1: No

Reviewer #3: **Yes: **Aaron A. Vogan

Reviewer #4: No

Figure Files:

Data Requirements:

Reproducibility:

References:

---

## [Editor Report · Decision Letter 2]

18 Jan 2023

Dear Prof Croll,

We are pleased to inform you that your manuscript 'Recent transposable element bursts are associated with the proximity to genes in a fungal plant pathogen' has been provisionally accepted for publication in PLOS Pathogens.

Best regards,

Bart Thomma

Section Editor

PLOS Pathogens

Bart Thomma

Section Editor

PLOS Pathogens

Kasturi Haldar

Editor-in-Chief

PLOS Pathogens

orcid.org/0000-0001-5065-158X

Michael Malim

Editor-in-Chief

PLOS Pathogens

orcid.org/0000-0002-7699-2064
---

## [Editor Report · Acceptance letter]

7 Feb 2023

Dear Prof Croll,

We are delighted to inform you that your manuscript, "Recent transposable element bursts are associated with the proximity to genes in a fungal plant pathogen," has been formally accepted for publication in PLOS Pathogens.

Best regards,

Kasturi Haldar

Editor-in-Chief

PLOS Pathogens

orcid.org/0000-0001-5065-158X

Michael Malim

Editor-in-Chief

PLOS Pathogens

orcid.org/0000-0002-7699-2064